# A fully coupled Arctic sea ice-ocean-atmosphere model (ArcIOAM v1.0) based on C-Coupler2: model description and preliminary results

Shihe Ren[1], Xi Liang[1], Qizhen Sun[1], Hao Yu[2], L. Bruno Tremblay[3], Bo Lin[1], Xiaoping Mai[1], Fu Zhao[1], Ming Li[1], Na Liu[1], Zhikun Chen[1], Yunfei Zhang[1]

[1]Key Laboratory of Research on Marine Hazards Forecasting, National Marine Environmental Forecasting Center, Ministry of Natural Resources, Beijing, China
[2]Ministry of Education Key Laboratory for Earth System Modelling, Department of Earth System Science, Tsinghua University, Beijing, China
[3]Department of Atmospheric and Oceanic Sciences, McGill University, Montreal, Canada

*Correspondence to*: Xi Liang (liangx@nmefc.cn)

**Abstract.** Aiming at reliable Arctic sea ice prediction on seasonal timescale in the National Marine Environmental Forecasting Center of China, a new Arctic regional coupled sea ice-ocean-atmosphere model (ArcIOAM) has been established. The description and implementation of ArcIOAM and its preliminary results in the year of 2012 are presented in this paper. In the ArcIOAM configuration, the Community Coupler 2 (C-Coupler2) is used to couple the Arctic sea ice-oceanic configuration of the MITgcm (Massachusetts Institute of Technology general circulation model) with the Arctic atmospheric configuration of the Polar WRF (Weather Research and Forecasting) model. A scalability test is performed to investigate the parallelization of the coupled model. As the first step toward reliable Arctic seasonal sea ice prediction, the simulation results in the year of 2012 of the ArcIOAM implemented with two-way coupling strategy, along with one-way coupling strategy, are evaluated with respect to available observational data and reanalysis products. Besides, the standalone MITgcm run with prescribed atmospheric forcing is performed for references. From the comparison, all the experiments simulate rational evolution of sea ice and ocean states in the Arctic region over a one-year simulation period. The two-way coupling strategy has better performance in terms of sea ice extent, concentration, thickness and SST, especially in summer. This result indicates that sea ice-ocean-atmosphere interaction takes a crucial role in controlling Arctic summertime sea ice distribution. The coupled model and documentation are available at https://doi.org/10.5281/zenodo.3742692 (last access: 9 June 2020), and the source code is maintained at https://github.com/cdmpbp123/Coupled_Atm_Ice_Oce (last access: 7 April 2020).

## 1 Introduction

It has been widely recognized that coupling between different earth system components (ocean, atmosphere, sea ice, and land) could provide improved forecasts of oceanic and atmospheric states on various timescales (Neelin et al., 1994). As an

essential component in climate system, sea ice plays a crucial role in global energy and water budget, and has a substantial impact on local and remote atmospheric and oceanic circulations. In polar region, strong interactions between different interfaces disturb sea ice motion and affect sea ice growth-melt process (Jung et al., 2016). Due to the combined features of solid and fluid, sea ice thermodynamical and dynamical representations in coupled models are complicated (Bailey et al.,
2020). In recent years, marine traffic through the Arctic marginal seas are projected to become increasingly feasible as climate change continues, which has amplified the demand for reliable polar sea ice and marine environmental predictions from synoptic timescale to seasonal and interannual timescales.

In the past decades, a number of coupled models have been developed with various sea ice prediction capacities on various timescales (Pellerin et al., 2004;Williams et al., 2018;Chen et al., 2010;Skachko et al., 2019). Climate models comprising
phase 6 of the Coupled Model Intercomparison Project (CMIP6) represent state-of-the-art sea ice prediction and outlook on seasonal to longer timescales. Recently within the GODAE (Global Ocean Data Assimilation Experiment) Oceanview community, there is an increasing interest of using coupled global models to predict sea ice on shorter timescales (Brassington et al., 2015). In Canada, a coupled global forecasting system is now operationally running at the Canadian Centre for Meteorological and Environmental Prediction (Smith et al., 2018), providing global 10 days forecasts of ocean
and sea ice states. The ocean-sea ice component of this system, namely the Global Ice-Ocean Prediction System (GIOPS, runs in real time since March 2014) (Smith et al., 2016), are based on the Nucleus for European Modelling of the Ocean (NEMO) and the Community Ice CodE (CICE) model. The GIOPS is coupled to an operational global deterministic medium-range weather forecasting system, namely the Global Deterministic Prediction System (GDPS) (Smith et al., 2014), which is based on the Global Environmental Multiscale (GEM) atmosphere model. In the United Kingdom, Hadley Centre
Global Environment Model version 3 (HadGEM3) is under development and is planning to service in seasonal sea ice prediction (Williams et al., 2018). The HadGEM3 is constitute of the UK Met Office Unified Model (UKMO UM) atmosphere model (Walters et al., 2011), the Joint UK Land Environment Simulator land-surface model (Brown et al., 2012), the NEMO model and the CICE model. In the United States, a coupled global sea ice-ocean-wave-land-atmosphere prediction system providing operational daily predictions out to 10 days and weekly predictions out to 30 days is being
developed by the US Navy (Brassington et al., 2015;Posey et al., 2015).

Although global coupled models are now being implemented with increased horizontal resolution, higher-resolution regional coupled models can provide an affordable way to study interactive ocean-atmosphere and sea ice-atmosphere feedbacks for polar weather and sea ice processes, if properly forced by initial and boundary conditions. On the regional scale, there are also a few coupled sea ice-ocean-atmosphere model systems for the Arctic climate studies and operational sea ice forecasts.
The Arctic Region Climate System Model (ARCSyM) was developed to simulate coupled interactions among the atmosphere, sea ice, ocean, and land surface of the western Arctic (Lynch et al., 1995;Rinke et al., 2000). Schrum et al. (2003) introduced a coupled sea ice-ocean-atmosphere model for the North and Baltic Seas. In their work, the regional atmospheric model REgional MOdel (REMO) was coupled to the HAMburg Shelf Ocean Model (HAMSOM) with a sea ice module. Pellerin et al. (2004) demonstrated that significant sea ice forecasting improvements occurred when implemented

the two-way coupling between the Gulf of St. Lawrence model with the GEM atmosphere model. The Regional Arctic System Model (RASM) is a fully coupled, regional Earth system model covering the pan-Arctic domain (Maslowski et al., 2012;Cassano et al., 2017). The component models of RASM include the Weather Research and Forecasting (WRF) atmospheric model, the Variable Infiltration Capacity (VIC) land and hydrology model, and regionally configured versions of the ocean and sea ice models used in the Community Earth System Model (CESM): the CICE model and Parallel Ocean

Program (POP). Van Pham et al. (2014) compared basin-scale climate simulation in the regional coupled model COSMO-CLM-NEMO with that in the stand-alone COSMO-CLM model for the North and Baltic Seas, and found large improvement in the simulated atmospheric low boundary temperature. As part of the Canadian Operational Network of Coupled Environmental PredicTion Systems (CONCEPTS), a fully coupled sea ice-ocean-atmosphere forecasting system for the Gulf of St. Lawrence has been developed (Faucher et al., 2009) and running operationally at the Canadian Meteorological Centre

since June 2011. The new model developing plan is to couple a high-resolution (1/12 degree) sea ice-ocean regional model which covering the North Atlantic and Arctic Ocean (Dupont et al., 2015) to the regional weather and wave prediction system of Environment Canada and provides short-term sea ice and ocean predictions to users. Yang et al. (2020) has developed a coupled atmosphere-seaice-ocean model configured for the pan-Arctic with the Coupled Ocean-Atmosphere-Wave-Sediment Transport modeling system (COAWST). A data assimilation system of ensemble Kalman filter is combined

with this coupled model to assimilate satellite sea ice observations to improve initial sea ice conditions. Since regional models can be run at higher resolution than global models, regional models can explicitly represent mesoscale features that may not be resolved in global models. Another potential advantage of regional systems is that lateral boundary conditions can be controlled to get an optimal model input (Cassano et al., 2017). In coupled model systems, moisture, heat and momentum are often accomplished through the use of a separate coupling software like OASIS-MCT (Craig et al., 2017) or

framework like the Earth System Model Framework (ESMF) (DeLuca et al., 2012) which links component models flexibly and controls the exchange and interpolation of coupling variables. The coupler, which can handle data interpolation and data transfer between different models and different grids, is the crucial part in the coupled systems. Using the ESMF and the National United Operational Prediction Capability (NUOPC), Sun et al. (2019) introduced a regional ocean-atmosphere coupled model covering the Red Sea based on the MITgcm (Marshall et al., 1997) and the WRF model (Skamarock et al.,

90  2008).

Aiming at providing operational seasonal sea ice prediction in the National Marine Environmental Forecasting Center (NMEFC) of China, the motivation of this work is to establish a fully coupled Arctic sea ice-ocean-atmosphere model (ArcIOAM) as a new tool to perform regional sea ice simulation and operational sea ice prediction on seasonal timescale.. In our study, we use a newly developed efficient coupling framework, the Community Coupler 2 (C-Coupler2) (Liu et al.,

2018), to couple the Arctic sea ice-oceanic configuration of the MITgcm (Nguyen et al., 2011;Liang and Losch, 2018) with the Arctic atmospheric configuration of the Polar WRF model (Hines and Bromwich, 2008) model. By coupling the Polar WRF and the MITgcm for the first time in Arctic region, a series of specific procedures including data interpolation between different grids and relaxation algorithm in lateral boundaries are implemented. After implementing the ArcIOAM, we

evaluate the model performance in the year of 2012 against available observational and reanalysis data. This year is selected

because of the historical sea ice extent minimum record in the satellite era. To evaluate the role of sea ice-ocean-atmosphere interaction in Arctic sea ice seasonal cycle, we compare the simulation results of the two-way coupling experiment with that of the one-way coupling experiment in which the coupling variables are only transmitted from the Polar WRF to the MITgcm. Besides, a stand-alone MITgcm simulation with prescribed atmospheric forcing is performed for references.

The paper is organized as follows. The description of the component models and coupling strategy are detailed in Section 2.

In section 3, a scalability test of the coupled model is performed to investigate its parallel capability. Section 4 introduces the designs and configurations of coupling experiments. Section 5 discusses the preliminary results in the validation test. The last section concludes the paper and presents an outlook for future work.

## 2 Model Description

The newly developed regional coupled modeling system of ArcIOAM is introduced in this section. The descriptions of

individual model components and the coupling strategy with C-Coupler2 are presented below. Detailed options of physical parameterizations and model settings for the Polar WRF, MITgcm models and C-Coupler2 are summarized in Table **1**.

### 2.1 The Oceanic and Sea Ice Component Model

The ocean and sea ice component of ArcIOAM is an Arctic configuration of the MITgcm (Nguyen et al., 2011;Liang and Losch, 2018;Liang et al., 2019;Liang et al., 2020). The model has an average horizontal resolution of 18 km and covers the

whole Arctic Ocean with open boundaries close to 55 °N in both the Atlantic and Pacific sectors (Losch et al., 2010). The ocean model includes 420x384 horizontal grid points and 50 vertical model layers based on Arakawa C grid and Z coordinates. The ocean model uses curvilinear coordinates and the model grid is locally orthogonal. Vertical resolution of the ocean model layers increases from 10 m near the surface to 456 m near the bottom. The K-profile parameterization (KPP) (Large et al., 1994) is used as the vertical mixing scheme. Time step is 1200 seconds.

The sea ice model shares the same horizontal grid with the ocean model and divides each model grid into two parts: ice and open ocean. In the open ocean area, ocean-atmosphere heat and momentum fluxes are calculated following the standard bulk formula (Doney et al., 1998). In the ice-covered area, the ice surface and bottom heat and momentum fluxes are calculated according to viscous-plastic dynamics and zero-layer thermodynamics (Hibler, 1980;Semtner, 1976). The so-called zero-layer thermodynamic model assumes one-layer ice underneath one-layer snow and ice does not store heat, therefore tends to

exaggerate the seasonal variability in ice thickness. Snow modifies ice surface albedo and conductivity. If enough snow accumulates on top of the ice, its weight submerges the ice and the snow is flooded. In order to parameterize a sub-grid scale distribution for sea ice thickness, the mean sea ice thickness in each grid can be split into as many as 7 thickness categories in the MITgcm sea ice model. In our coupled model for simplicity, we use 2 thickness categories: open water and sea ice.

## 2.2 The Atmospheric Component Model

The atmospheric component of ArcIOAM is based on an Arctic configuration of the Polar WRF (Bromwich et al., 2013;Hines and Bromwich, 2008) model, which is an optimized version of the WRF model (Skamarock et al., 2008) for use in polar region. The Polar WRF is developed and maintained by the Polar Meteorology Group at the Byrd Polar and Climate Research Center of the Ohio State University. In the Arctic configuration of the Polar WRF model, modifications for polar environments primarily encompass the land surface model and sea ice to adapt to the particular conditions in Arctic Regions.

The Noah land surface model is embedded inside the Polar WRF. The changes made in the Noah land surface model (LSM; Chen and Dudhia, 2001) include using the latent heat of sublimation for calculating latent heat flux over ice surface, increasing the snow albedo and the emissivity value for snow, adjusting snow density, modifying thermal diffusivity and snow heat capacity for the subsurface layer, changing the calculation of skin temperature, and assuming ice saturation in calculating the surface saturation mixing ratio over ice. Other modifications for the Polar WRF include a fix to allow

specified sea ice quantities and the land mask associated with sea ice to update during a simulation. These modifications improve model performance over the pan-Arctic for short-term forecasts.

The Arctic configuration of the Polar WRF model has been tested and evaluated by a set of simulations over several key surface categories, including large permanent ice sheets with the Greenland/North Atlantic grid and Arctic land (Hines et al., 2011;Hines and Bromwich, 2008) and the production of the Arctic System Reanalysis (ASR) (Bromwich et al., 2010). In this

study, the Polar WRF model covers the Arctic regions with a horizontal resolution of 27 km. The model has 306x306 horizontal grid points and 60 vertical layers. The prognostic equations in the Polar WRF model are solved with a time step of 120 seconds. The Polar WRF model employed physics options that included the Mellor Yamada-Janjic boundary layer scheme in conjunction with the Janjic-Eta Monin Obukhov surface layer scheme (Janjić, 2002), the WRF single-moment 6-class microphysics scheme for microphysics, the Grell-Devenyi scheme for clouds (Grell and Dévényi, 2002), and the new

version of the rapid radiative transfer model for both shortwave and longwave radiation.

## 2.3 The Coupler

We have implemented the C-Coupler2 to couple the MITgcm and the Polar WRF model. The C-Coupler family was initiated from 2010 in China. The first version (C-Coupler1) includes features such as flexible coupling configuration and 3-D coupling capability (Liu et al., 2014). Two coupled models have been built using the C-Coupler1. The first is a coupled

climate system model version FGOALS-gc at the Institute of Atmospheric Physics, Chinese Academy of Sciences. The FGOALS-gc can achieve exactly the same (bitwise identical) simulation results as same model components with different coupler the CPL6 (Liu et al., 2014). The second is a regional coupled model FIO-AOW (Zhao et al., 2017) which includes an atmosphere model WRF, an ocean model POM (Princeton Ocean Model) , and a wave model MASNUM (Yang et al., 2005). The second version of the C-Coupler family, the C-Coupler2 (Liu et al., 2018), is equipped with many advanced functions,

including 1) a common, flexible, user-friendly coupling configuration interface, 2) the capability of coupling within one

executable or the same subset of Message Passing Interface (MPI) processes, 3) flexible and automatic coupling procedure generation for any subset of component models, 4) dynamic 3-D coupling that enables convenient coupling of field on 3-D grids with time-evolving vertical coordinate values, 5) non-blocking data transfer, 6) facilitation for model nesting, 7) facilitation for increment coupling and 8) adaptive restart capability (Liu et al., 2018).

## 2.4 Coupling Strategy

In the ArcIOAM, the requested CPUs are assigned equally to the MITgcm and Polar WRF model. The C-Coupler2 is employed as a library to achieve the two-way parallel coupling between the Polar WRF and the MITgcm (Figure 1). The coupling interval is set to 20 minutes. The component models are running in concurrent mode (Figure 2), that is, the component models run on mutually exclusive sets of cores, if one component model finishes earlier than the other, its resources are idle and wait for the other component model. At each coupling time step, data transfer from the MITgcm to the Polar WRF is executed when data transfer from the Polar WRF to the MITgcm is completed, and vice versa for choice. During coupling execution, the MITgcm sends SST, sea ice concentration, sea ice thickness, snow depth and ice surface albedo to the coupler, and these coupling variables are used directly as the bottom boundary conditions in the Polar WRF model. The Polar WRF model sends the atmospheric bottom boundary variables to the coupler, including downward longwave radiation, downward shortwave radiation, 10-m wind speed, 2-m air temperature, 2-m air specific humidity, and precipitation. The MITgcm uses these atmospheric variables to compute the open ocean and ice surface heat, freshwater and momentum forcing.

Model domain of the MITgcm and the Polar WRF model are shown in Figure 3a. As the model domain and grid of the Polar WRF and the MITgcm are generally different, several important procedures have been carried out in conducting our coupled system. The model domain of the Polar WRF is larger than that of the MITgcm, producing a non-overlapped area between the MITgcm domain and the Polar WRF domain. Besides, the MITgcm model only produces surface variables over ocean, and the Polar WRF model also needs bottom boundary conditions over land. Thus, the coupling variables received by the Polar WRF model need to be concatenated by value in the non-overlapped area and in the land area from an external forcing file, and value in the overlapped ocean area from the MITgcm model together. To diminish the abrupt value changes from two sources, a simple linear relax zone is designed near the open boundaries of the MITgcm model in both the Atlantic and Pacific sectors (Figure 3b). The coupling variables ($VAR_{recbyWRF}$) received by the Polar WRF model can be expressed as:

$$VAR_{recbyWRF} = (1 - \alpha)VAR_{sedbyMIT} + \alpha VAR_{extern} \qquad (1)$$

where $\alpha$ is relaxation coefficient, which is equal to 0 in the overlapped ocean area away from the MITgcm open boundaries, and equal to 1 in the land area and in the non-overlapped area away from the MITgcm open boundaries. While in the relax zone, $\alpha$ increases from 0 to 1 linearly from the overlapped side to the non-overlapped side. $VAR_{sedbyMIT}$ is the coupling variables which are send by the MITgcm model. $VAR_{extern}$ is the bottom boundary variables of the Polar WRF model which are read from external forcing file.

Normally in coupled models the coupler controls the exchange of heat and momentum fluxes among component models. In our model configuration, instead of coupling fluxes directly, we use the C-Coupler2 to control the exchange of fields between the Polar WRF and the MITgcm. Heat and momentum fluxes are calculated separately in each component model. Both the Polar WRF and the MITgcm use the same Bulk Formula and almost same parameters in calculating fluxes, which guarantees the quasi-conservation of heat and momentum transmission between the component models. The bilinear interpolation algorithm is involved in the transmission of model variables between the horizontal grid of the Polar WRF and that of the MITgcm. Figure 4shows wind stress curl derived from the Polar WRF output and the MITgcm output, as well as their difference on March 1, 2012. It can be seen that the Polar WRF and MITgcm model generate similar wind stress curl pattern, and the difference due to interpolation algorithm and momentum calculation accounts for less than 5% of the wind stress curl (Figure 4c).

## 3 Scalability test

In this section, the parallel efficiency of the ArcIOAM is investigated. Different numbers of CPU cores are used to evaluate the parallel speed-up of the coupled model. The CPU elapsed time spent on coupling interface of each component model in the coupled runs are detailed. Additionally, the parallel efficiency of each component model in the stand-alone runs are calculated for references. The parallel efficiency tests are performed on the High performance computing cluster at NMEFC. The High performance computing cluster is a Lenovo Blade Server system composed of 240 dual-socket compute nodes based on 14-core Intel Haswell processors running at 2.4 GHz. Each node has 128GB DDR4 memory running at 2133 MHz. Overall the system has a total of 6270 CPU cores (240 nodes x 2 x 14 CPU cores) and has a theoretical peak speed of 258 tetaflops. The parallel efficiency of the scalability test is $N_{p0} t_{p0}/ N_{pn} t_{pn}$, where $N_{p0}$ and $N_{pn}$ are the number of CPUs employed in the base case and the test case, respectively; $t_{p0}$ and $t_{pn}$ represent the CPU elapsed time in the base case and the test case. The speed-up is defined as $t_{p0} / t_{pn}$, which is the relative improvement of the CPU time. The scalability tests are performed by integrating 7 model days for the stand-alone Polar WRF, the stand-alone MITgcm and the coupled runs.

In the ArcIOAM runs, the requested CPUs are assigned equally to the component models. The minimum CPUs we use is 28, i. e. $N_{p0} = 28$. Limited by computational resource, the maximum CPUs we can use is 896. The total CPU elapsed time in the coupled runs decreases from 12840 s to 1380 s when the requested CPUs increases from 28 to 896 (Table 2). When the requested CPUs are not larger than 448, the CPU elapsed time used for numerical integration by the MITgcm is substantially smaller than that for numerical integration by the WRF, meaning that the efficiency of the coupled model depends on the WRF component model. When the requested CPUs are larger than 448, the efficiency of the coupled model depends on the MITgcm component model.

The parallel efficiency of the coupled model remains more than 90% when employing less than 112 cores and is still as high as 80% when using 224 cores (Figure 5). The  parallel efficiency of the stand-alone MITgcm is near to that of the stand-alone Polar WRF when the requested CPUs are less than 448, while both of them are substantially lower than the coupled

model. The parallel speed-up of the coupled model is higher than the stand-alone component model. The decrease in parallel efficiency results from the increase of communication time, load imbalance, and I/O (read and write) operation per CPU core (Christidis, 2015).

## 4 Numerical Experiments

The ArcIOAM will be used to conduct seasonal sea ice prediction in our future plan. As a starting point, we need to evaluate
its performance on seasonal timescale. In this work, we perform the coupled model simulations in the year of 2012. The year of 2012 is chosen because an unusually strong storm formed off the coast of Alaska on 5 August 2012, and tracked into the center of the Arctic Basin where it lingered for several days and generated stronger sea ice-ocean-atmosphere interaction (Simmonds and Rudeva, 2012). With more open ocean area be exposed to atmosphere, we expect that sea ice-ocean-atmosphere interaction processes are relatively more intensified in the summertime than that in the wintertime. In the Arctic
region, demands of seasonal prediction for sea ice and ocean are also strong in summertime when more commercial and scientific activities of Arctic shipping occur. The main aim of this paper is to assess the sea ice and ocean simulation capabilities of the coupled system. For this reason, less attention will be paid on the atmosphere simulation. Future work will emphasize atmospheric variables and seasonal sea ice prediction skill with available observations be assimilated.

Three experiments using different coupling strategy are performed in this study (Table 3). The first experiment which
denoted by OCNCPL is a two-way coupled simulation that the MITgcm receives the coupled variables from the Polar WRF, and the Polar WRF also receives the coupled variables from the MITgcm. The second experiment which denoted by OCNDYN is a one-way coupled simulation that the MITgcm only receives the coupled variables from the Polar WRF, but without sending the coupled variables back to the Polar WRF. α in Equ. 1 is set to 1 in the OCNDYN run. The third experiment of OCNSTA represents the stand-alone MITgcm simulation with the same sea ice albedo parameters to the
coupled model but prescribed atmospheric forcing to keep consistency with previous two coupling experiments. The model state deviation between these cases represents the influences of sea ice-ocean-atmosphere interaction in the Arctic Ocean.

The atmospheric initial and lateral boundary conditions, as well as bottom boundary conditions in the external forcing file used in the OCNCPL and OCNDYN runs, and the prescribed atmospheric forcing used in the OCNSTA run are derived from the 6-hourly National Centers for Environmental Prediction (NCEP) Climate Forecast System Reanalysis (CFSR) data (Saha
et al., 2010). The oceanic monthly lateral boundary condition of the coupled model is derived from the Estimating the Circulation and Climate of the Ocean phase II (ECCO2): high-resolution global-ocean and sea ice data synthesis (Menemenlis et al., 2008), including potential temperature, salinity, current, and sea surface elevation. The discrepancy of atmosphere and ocean boundary condition is less of an issue since the ocean does not vary much on shorter time scale and the zones of sea ice are far away from the lateral boundary. The initial condition of ocean and sea ice on 1 January 2012 are
derived from a stand-alone MITgcm simulation initialized from climatological temperature and salinity field derived from the World Ocean Atlas 2005 (WOA05) (Locarnini et al., 2006;Antonov et al., 2006) and forced by the 3-hourly Japanese 55-

year Reanalysis data (JRA55) (Harada et al., 2016;Kobayashi et al., 2015) from 1979 to 2011 (Liang and Losch, 2018). After 33-year integration, the ocean and sea ice initial condition on 1 January 2012 used in the coupled model are retrieved from a quasi-equilibrium ocean-sea ice evolution period. River runoff is based on the Arctic Runoff Data Base (Nguyen et al., 2011). The model states are outputted on daily basis.

## 5 Preliminary Results

### 5.1 Sea Ice Extent and Concentration

The Arctic sea ice extent minimum value appeared in the summer of 2012 in the satellite era (Francis, 2013). According to sea ice extent record derived from the Multisensor Analyzed Sea Ice Extent-North Hemisphere (MASIE-NH) (NSIDC, 2010), Arctic sea ice extent grows to maximum value of 14.5 million $km^2$ in March and drops to minimum value of 3.5 million $km^2$ in September (Figure 6a) in the year of 2012. The MASIE-NH data is provided daily by the National Ice Center Interactive Multisensor Snow and Ice Mapping System with a spatial resolution of 4 km. Compared with the OCNSTA run, results from the experiments with coupling (OCNCPL and OCNDYN) are closer to observations. It is noted that both the OCNCPL and OCNDYN runs simulate lower sea ice extent than the observations by a bias of 1-1.5 million $km^2$ (Figure 6a) after the first half month of January. Because sea ice initial field on 1 January 2012 is derived from a stand-alone MITgcm simulation which is forced by the JRA55 data, the change of atmospheric forcing data from the JRA55 to the NCEP CFSR induces a model state adjustment period which lasts about half month. By comparing the sea ice extent evolution of the OCNCPL and OCNDYN run, it seems that sea ice-ocean-atmosphere interaction generates slight sea ice extent change, but based on our following analysis related to sea ice spatial distribution, sea ice-ocean-atmosphere interaction plays a decisive role in summertime sea ice spatial distribution.

Figure 6b shows the modeled and observed sea ice extent anomaly. After the model state adjustment period, both the amplitudes and phase of sea ice extent seasonal cycle in the OCNCPL and OCNDYN runs are close to the observations. While results of stand-alone run shows lag of sea ice melts and freezes in advance compared with the observations. Nguyen et al. (2011) pointed out that optimized parameters of sea ice and snow albedo depend on selected atmospheric forcing in the MITgcm. In the sea ice model of MITgcm, the actual surface albedo changes with time and is a function of four foundational albedo parameters (dry ice, dry snow, wet ice, wet snow), as well as ice surface temperature and snow depth. A series of sensitivitiy experiments are performed to get an optimal combination of sea ice parameters (figures not shown). The sea ice model systematic bias could also be reduced by involving sea ice data assimilation module (Liang et al., 2019) when conducting seasonal sea ice prediction system.

The modeled sea ice concentration is compared with the observations derived from the EUMETSAT Ocean and Sea Ice Satellite Application Facility (OSISAF) (Eastwood et al., 2011). The observations are reprocessed daily sea ice concentration fields which are retrieved from the Scanning Multichannel Microwave Radiometer/Special Sensor Microwave Imager (SMMR/SSMI) data with a spatial resolution of 10 km. Figure 6c shows the root mean square error (RMSE) evolution of the

modeled sea ice concentration with respect to the OSISAF data. After 1 month of model state adjustment, three experiments

shows similar patterns that RMSE is lower in winter and spring than in summer and autumn. The Arctic basin is almost fully covered by sea ice from November to May (Figure 7), thus the two coupling experiments do not produce substantial sea ice concentration differences. Along with more open ocean are exposed to atmosphere, from June to September the sea ice concentration RMSE of the OCNCPL run is significantly lower than that of the OCNDYN run. This result indicates that sea ice-ocean-atmosphere interaction takes a crucial role in controlling Arctic summertime sea ice distribution.

To further clarify sea ice concentration spatial distribution, we show the modeled and observed monthly mean sea ice concentration (Figure 7) and deviation of model results and observation (Figure 8) in March, June, September and December In March when the Arctic Ocean is almost fully covered by sea ice, the main source of discrepancy appears in sea ice edge zones in the Atlantic side (Figure 7a-c). In June, sea ice concentrations are overestimated in the Arctic marginal seas in the OCNCPL and OCNDYN runs (Figure 8d-e). The modeled sea ice concentration in the OCNSTA run is more closer to the

observations (Figure 7f). In September, the modeled sea ice in the marginal sea ice zone melts out in all runs (Figure 7i-k). Compared with the satellite observations (Figure 7l), sea ice in the OCNSTA run overmelts in summertime which leads to an anomalous negative bias of sea ice concentration in the Arctic (Figure 8i), the two coupled runs overestimate sea ice concentration in the southern Beaufort Sea while underestimates sea ice concentration in the center Arctic basin (Figure 8g-h). Although the two coupled runs simulate similar sea ice extent patterns, due to rational representation of sea ice-ocean-

atmosphere interaction in the OCNCPL run, the modeled sea ice distribution of the OCNCPL run is closer to the observations (Figure 7i and Figure 7l). In December, the situation is similar with that in March when sea ice dominates almost entire Arctic region.

## 5.2 Sea Ice Volume and Thickness

At current stage, satellite sea ice thickness data is not available in melting seasons from May to September. We compare the

modeled sea ice volume with that from a widely used sea ice volume data source (Figure 9a), the Pan-Arctic Ice Ocean Modeling and Assimilation System (PIOMAS) developed at the Applied Physics Laboratory of the University of Washington (Zhang and Rothrock, 2003). The PIOMAS assimilates sea ice concentration data from the National Snow and Ice Data Center (NSIDC) and SST data from NCEP/NCAR Reanalysis. The OCNSTA run simulates more rational sea ice growth rate from January to May but systematic negative sea ice volume bias compared with the PIOMAS data. The sea ice

volume in the OCNCPL and OCNDYN runs shows better results than that in the OCNSTA run from June to December. However, both the two coupled runs produce less sea ice volume than the PIOMAS data in most time of 2012, partly resulting from that our model underestimates sea ice extent (Figure 6a) without assimilating any observation. It is notable that the sea ice volume evolution of the OCNCPL run is closer to the PIOMAS data at the end of 2012.

Satellite sea ice thickness observations are usually retrieved from either ice surface brightness temperature or radar altimetric

measurement of sea ice freeboard. We use three kinds of satellite sea ice thickness data to validate our model results (Figure 9b and Figure 9c). Daily sea ice thickness observations provided by the University of Hamburg are derived from the Soil

Moisture Ocean Salinity (SMOS) brightness temperature combined with a sea ice thermodynamic model and a three-layer radiative transfer model (Kaleschke et al., 2012) obtained from http://icdc.cen.uni- hamburg.de/1/daten/cryosphere/l3c-smos-sit.html. Weekly sea ice thickness observations provided by the Alfred Wegener Institute, Helmholtz Centre for Polar and Marine Research are derived from the European Space Agency satellite mission CryoSat-2 radar altimetric data (Ricker et al., 2014) obtained from http://data.meereisportal.de/data/cryosat2/version2.0/. The SMOS observations retrieved from satellite brightness temperature data have promised qualities in marginal sea ice zone where ice thickness is thinner than 1 m (Tian-Kunze et al., 2014) while the CryoSat-2 observations retrieved from radar altimetric data have higher accuracies in pack sea ice zone than in marginal sea ice zone (Laxon et al., 2013;Wingham et al., 2006). Taking the spatial complementarity of the SMOS and CryoSat-2 data into consideration, Ricker et al. (2017) introduced a weekly sea ice thickness product covering the entire Arctic, the CS2SMOS sea ice thickness, which is generated by merging the SMOS sea ice thickness with the CryoSat-2 sea ice thickness (Ricker et al., 2017) obtained from https://data.meereisportal.de/data/cs2smos/version1.4/. The CS2SMOS data with observational uncertainty is also added in the comparison.

The weekly CryoSat-2 data is constitute of several banded sea ice thickness records which collected in one week when polar orbital satellite passes the Arctic region. The SMOS data used in this study are those in thin ice (< 1 m) region. Considering spatial coverage of the observations, we compare spatial-mean sea ice thickness evolution with the CS2SMOS data (Figure 9b). Comparing with the CS2SMOS data, both coupled runs produce more rational sea ice thickness evolution than stand-alone run from January to April. However, large sea ice thickness errors between the model and the observations exist in October and November. We attribute these large errors to the possibly observational uncertainties induced by radar altimetric measurement errors when ice surface starts to freeze up. The modeled sea ice in the OCNCPL run is thinner than that in the OCNDYN run, and the sea ice thickness deviations between the two runs amplify after the summer. Meanwhile the sea ice volume and thickness of the OCNCPL run are closer to the PIOMAS data and the CS2SMOS observations at the end of 2012. Day et al. (2014) pointed out that sea ice behaves long-term memory of melting-freezing processes. Notz and Bitz (2017) indicated that summertime sea ice thickness has an important influence on sea ice state in the following spring through the ice thickness-ice growth feedback. A negative anomaly of sea-ice area in late summer induces larger heat losses in autumn and winter from the ocean to the atmosphere due to enhanced outgoing long-wave radiation and turbulent heat fluxes, this causes thinner snow and ice due to later freeze-up and hence larger heat-conduction fluxes through sea ice, eventually leading to larger ice-growth rates. We speculate that in the OCNCPL run sea ice-ocean-atmosphere interaction induces reasonable sea ice thickness distribution in the summer of 2012 which preconditions the sea ice thickness evolution in the following freezing season.

The sea ice thickness RMSEs of the three runs with respect to mentioned three kinds of satellite sea ice thickness data are shown in Figure 9c. Compared with the coupled runs, the sea ice thickness in the OCNSTA run shows larger bias in pack ice zone while smaller bias in marginal ice zone. The sea ice thickness RMSE between the OCNCPL run and the SMOS data is smaller than that between the OCNDYN run and the SMOS data, indicating that sea ice-ocean-atmosphere interaction substantially improves the sea ice thickness simulation in the marginal sea ice zone in the coupled runs. The sea ice thickness

RMSEs between the coupled runs and the CryoSat-2 data are generally larger than those between the coupled runs and the CS2SMOS data especially in October and November, which is partly due to the large uncertainty of radar altimetric measurement when ice surface starts to freeze up, and partly due to the low spatial coverage of the CryoSat-2 data.

Normally satellite sea ice thickness data has large uncertainty due to limitation of retrieval algorithm. In situ sea ice thickness observations with higher accuracy can provide a direct reference for the model. To further evaluate the modeled sea ice thickness, we compare the time evolution of modeled and observed sea ice thickness at three locations in the Beaufort Sea in 2012 (Figure 10). The observations are derived from moored upward-looking sonar (ULS) ice draft data from the Beaufort Gyre Exploration Project (BGEP) (Proshutinsky et al., 2005). The ULS samples the ice draft with a precision of 0.1 m (Melling and Riedel, 1995), and the ice draft can be converted to ice thickness following the law of hydrostatic equilibrium (Nguyen et al., 2011). Generally speaking, at all three locations in the Beaufort Sea, when the modeled sea ice is thinner than 1 m, the sea ice thickness evolution improves in the OCNCPL run comparing with those in the OCNDYN run. This result further demonstrates that sea ice-ocean-atmosphere interaction plays an important role in marginal sea ice evolution.

Spatial distributions of monthly mean sea ice thickness and its bias with respect to available CS2SMOS data in March, June, September, and December are shown in Figure 11 and Figure 12. In March and December, all three runs underestimate sea ice thickness in central Arctic, while overestimate sea ice thickness in marginal sea ice zone (Figure 12). In March, the OCNSTA run overestimates sea ice thickness in the Pacific sector of the Arctic Ocean and in the Baffin Bay (Figure 12e). The coupled runs overestimate sea ice thickness in the northern Barents Sea while underestimate sea ice thickness in the western Chukchi Sea (Figure 12a and Figure 12c). In December, compared with the OCNDYN run, the modeled sea ice thickness in marginal sea ice zone in the OCNCPL run is more closer to the CS2SMOS data (Figure 12b), partly due to the rational sea ice distribution at the beginning of freezing season, as summertime sea ice thickness has strong effect on preconditioning the following wintertime sea ice thickness (Day et al., 2014).

## 5.3 Ocean Temperature and Current

Sea ice states are intimately linked to ocean states, both dynamically and thermodynamically. The modeled spatial distribution of sea ice concentration in the OCNCPL run exhibits great improvement comparing with the OCNDYN run. Since sea ice in marginal ice zone is strongly affected by SST through lateral heat transport, we suspect that sea ice-ocean-atmosphere interaction should impose positive influence on the modeled ocean temperature in the marginal sea ice zone.

The modeled SST is validated against the Group for High-Resolution SST Multi-Product Ensemble (GMPE) data (obtained from http://marine.copernicus.eu/, product identifier: SST_GLO_SST_L4_NRT_OBSERVATIONS_010_005). The GMPE SST data provided by the UKMO is a reanalysis daily global SST product that computed as the median of a large number of SST products by various institutes around the world. Each product contributing to the GMPE product uses different observational data sets or different retrieval algorithms. As a median product of multiproduct ensemble, the GMPE SST data greatly reduces observational uncertainties. The SST RMSE of the three runs with respect to the GMPE data are shown in

Figure 13. In general compared with the coupled runs, the SST RMSE in the OCNSTA run is smaller in the summertime but
larger in the rest. Spatial patterns of the modeled and observed SST in March, June, September and December are shown in
Figure 14. Deviation of the modeled SST and the GMPE SST observation is demonstrated in Figure 15. The GMPE SST
data is available in ice-free areas (Figure 14d, Figure 14h, Figure 14l and Figure 14p). In March and June, the OCNSTA run
produces a warmer sea surface in the Nordic Seas, which explains the positive SST bias from January to June in Figure 13
compared with the coupled runs. In September the SST RMSE in the OCNCPL run (Figure 13) arises from the strong
negative bias in the southern Beaufort Sea and the Baffin Bay (Figure 15g).

Ocean current observations in the Arctic Ocean are quite sparse, we evaluate the modeled ocean velocity and temperature
with climatological observation generated from the 1998-2003 mooring data in Fram Strait. Under the framework of the
European Union projects Variability of Exchanges In the Northern Seas (VEINS) and Arctic Subarctic Ocean Fluxes - North
(ASOF-N), a series of moorings in Fram Strait had been deployed to record ocean properties since September 1997 to 2004
(obtained from https://www.whoi.edu/page.do?pid=30914). The observation covers water column from 10 m above the
seabed to about 50 m below the surface. Although the observations were conducted at least one decade earlier than 2012, we
believe that the comparison between the modeled and observed monthly mean value would likely still make sense since the
phase of the Atlantic Multidecadal Oscillation does not reverse between 1995 and 2012. The modeled and observed
northward cross-section velocity and temperature averaged between 5°E and 8°40'E at 78°50'N are listed in Table 4.
Basically, the observations show that the northward velocity of the West Spitsbergen Current (WSC) increases from July to
September, and the mean temperature of the section of 78°50'N also increases from July to December. It is notable that the
modeled velocity and temperature of the OCNCPL run in Fram Strait are closer to the observations comparing with those of
the OCNDYN run, although there are still large biases of the modeled velocity between the OCNCPL run and the
observations. Vertical temperature distribution in the section averaged between July and September shows that sea ice-
ocean-atmosphere interaction induces warming of the WSC until 700 m depth accompanied with strong cooling beside the
WSC (Figure 16c). The cross-section velocity deviation between the OCNCPL and OCNDYN run is characterized by
enhanced northward velocity over the whole water column around 0 °E and east of 6 °E, while reduced northward velocity
between them (Figure 16f).

## 6 Conclusion and Discussion

This paper describes the implementation of an Arctic regional sea ice-ocean-atmosphere coupled model (ArcIOAM). To
connect the component models, a newly developed coupler, C-Coupler2 is implemented to couple the Arctic sea ice-oceanic
configuration of the MITgcm model with the Arctic atmospheric configuration of the Polar WRF model. By coupling the
Polar WRF and the MITgcm for the first time in Arctic region, a series of specific procedures including data interpolation
between different grids and relaxation algorithm in lateral boundaries are designed. The parallel efficiency of the coupled
model is also investigated.

After implementing the ArcIOAM, we demonstrate it on seasonal simulation of the Arctic sea ice and ocean states in 2012 to evaluate the model capability of seasonal prediction of sea ice. Results from the two-way coupling simulation (OCNCPL), the one-way coupling simulation (OCNDYN) and stand-alone oceanic simulation (OCNSTA) are compared to a wide variety of available observational and reanalysis products. The model state deviation between the two coupled experiments represents the influences of sea ice-ocean-atmosphere interaction on the Arctic Ocean and sea ice. From the comparison, results obtained from the two-way coupling experiment capture the sea ice and ocean evolution in the Arctic region over a 1-year simulation period. The two-way coupling experiment gives better results compared with the one-way coupling experiment and stand-alone oceanic simulation, especially in summertime.

Both the amplitudes of sea ice extent seasonal cycle of the two coupled runs are close to the observations. The spatial distribution of sea ice concentration in the OCNCPL run is similar to that in the OCNDYN run from January to May. From June to September the sea ice concentration RMSE of the OCNCPL run with respect to the observations is significantly lower than that of the OCNDYN run, indicating that sea ice-ocean-atmosphere interaction takes a crucial role in controlling Arctic summertime sea ice distribution. The sea ice thickness RMSE of the OCNCPL run with respect to the SMOS data in thin ice areas is smaller than that of the OCNDYN run. Meanwhile, the evolution of the modeled and observed sea ice thickness at three locations in the Beaufort Sea show that the modeled sea ice thickness evolution improves in the OCNCPL run when the ice is thinner than 1m. This result means that sea ice-ocean-atmosphere interaction is very likely to improve the sea ice thickness simulation in the marginal sea ice zone when considering feedback of ocean to atmosphere. Based on comparison with a series of mooring data in Fram Strait, the modeled velocity and temperature in the OCNCPL run are closer to the observations than those in the OCNDYN run, although large biases of the modeled velocity still exist. Comparing with the satellite data, the SST obtained in the OCNCPL run is also better than that in the OCNDYN run in summer 2012. The two-way coupling between the Polar WRF and the MITgcm provides a more rational representation of real air-ice-ocean physical processes, which includes the important ice-albedo feedback in early summer. In the MITgcm, sea ice albedo is calculated based on several variables, such as snow depth on ice, ice surface temperature. In the OCNCPL run, albedo is a coupling variable which affects both the Polar WRF and the MITgcm. In the OCNDYN run, albedo used in the Polar WRF is directly read from the CFSR forcing data. Due to strong sea ice-ocean-atmosphere interaction in summertime, the two-way coupling strategy not only improves the sea ice simulation, but also benefit the modeled ocean states.

The ArcIOAM is designed for seasonal sea ice prediction up to 6 months, while on longer timescale the regional model's capacity is expected to severely depend on the lateral boundary forcing data. Global coupled models, such as those involved in CMIP6, have innate advantages in sea ice prediction and outlook on seasonal to longer timescale because that the interactions between high- and mid- latitudes are considered. Land component is absolutely important to the Arctic simulation, however at current stage, our coupled model has not the capacity of coupling an individual land model, instead, we use the embedded land component in the Polar WRF for technical simplicity. It is noticed that the simulation presented in this paper only covers one year, more results for different years should be carried out to further assess the coupled model. However, given the encouraging results in 2012, this new developed Arctic regional coupled model exhibits its potential

capacity of seasonal sea ice prediction and provides a reliable basis for investigating both thermodynamic and dynamic process and forecasting applications in the Arctic sea ice scope. Meanwhile, bias in the modeled sea ice extent and summertime sea ice thickness still exist, although satellite sea thickness data normally has large uncertainty in summertime, which partly contributes to the large sea ice thickness bias in October-November between the model and CS2SMOS data (Figure 9b), the foundational sea ice albedo parameters in our current model configuration seem to be underestimated, which allows more heat into the ice and causes thinner sea ice thickness, as well as lower sea ice extent. The choice of sea ice albedo parameters also contributes to the large sea ice thickness bias in October-November between the model and CS2SMOS data. It is noticed that the albedo formulation in the MITgcm sea ice model is simple and straightforward. The CICE model provides a more sophisticated scheme for sea ice albedo calculation. On the way to operational seasonal sea ice prediction, the model physics and model uncertainty representation in the coupled model can be enhanced using advanced techniques, such as sophisticated sea ice albedo formulation, stochastic physics parameterizations and ensemble approaches. The regional coupled forecasting system also can be improved by involving data assimilation capabilities for initializing the forecasts. Future work will involve exploring these and other aspects for a regional coupled modeling system suited for forecasting and better understanding of mechanism.

*Code and data availability.* The latest version and future updates of the source code, user guide, and examples can be downloaded from https://github.com/cdmpbp123/Coupled_Atm_Ice_Oce (last access: 7 April 2020). The current version of this coupled model (ArcIOAM v1.0) used to produce the results in this paper can be accessed via https://doi.org/10.5281/zenodo.3742692.

*Author contribution.* SR and HY worked on the coding tasks for coupling the Polar WRF with the MITgcm using C-Coupler2. SR and XL designed and performed the simulations for the numerical experiments. SR, HY and XM worked on the technical details for debugging the model and wrote the code documentation. XL worked on the MITgcm model setup and performed sea ice analysis and validation. QS worked on the Polar WRF model setup. BL worked on the scalability test. All authors contributed to the writing of the final article.

*Competing interests.* The authors declare that they have no conflict of interest.

*Acknowledgments.* This work is supported by the National Key R&D Program of China (2016YFC1402700, 2018YFC1407200) and the National Natural Science Foundation of China (41806003). The authors thank the University of

Hamburg for providing the SMOS sea ice thickness data, the Alfred-Wegener-Institut, Helmholtz Zentrum für Polar- und Meeresforschung for providing the CryoSat-2, CS2SMOS sea ice thickness data and Fram Strait mooring data, the Norwegian Meteorological Institute for the OSISAF sea ice concentration data, the University of Washington for providing the PIOMAS sea ice volume data, the National Snow and Ice Data Center for providing the MASIE-NH data (http://nsidc.org/data/masie/), the Woods Hole Oceanographic Institution for providing the BGEP ULS data (http://www.whoi.edu/beaufortgyre), and the Copernicus Marine Environment Monitoring Service for providing the GMPE SST data (http://marine.copernicus.eu/).

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

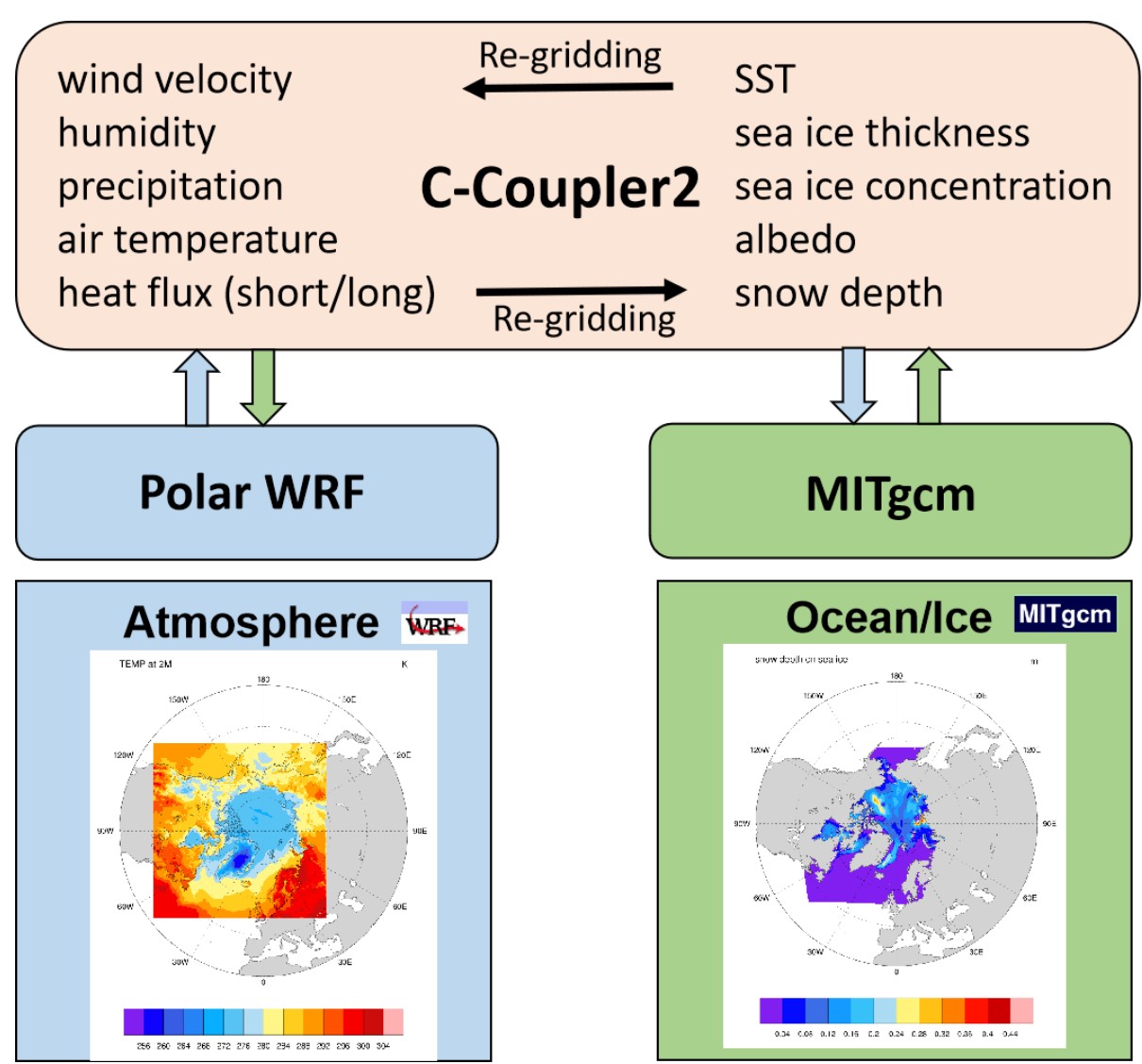

Figure 1: Coupling strategy of the Polar WRF-MITgcm coupled model system.

# Concurrent mode

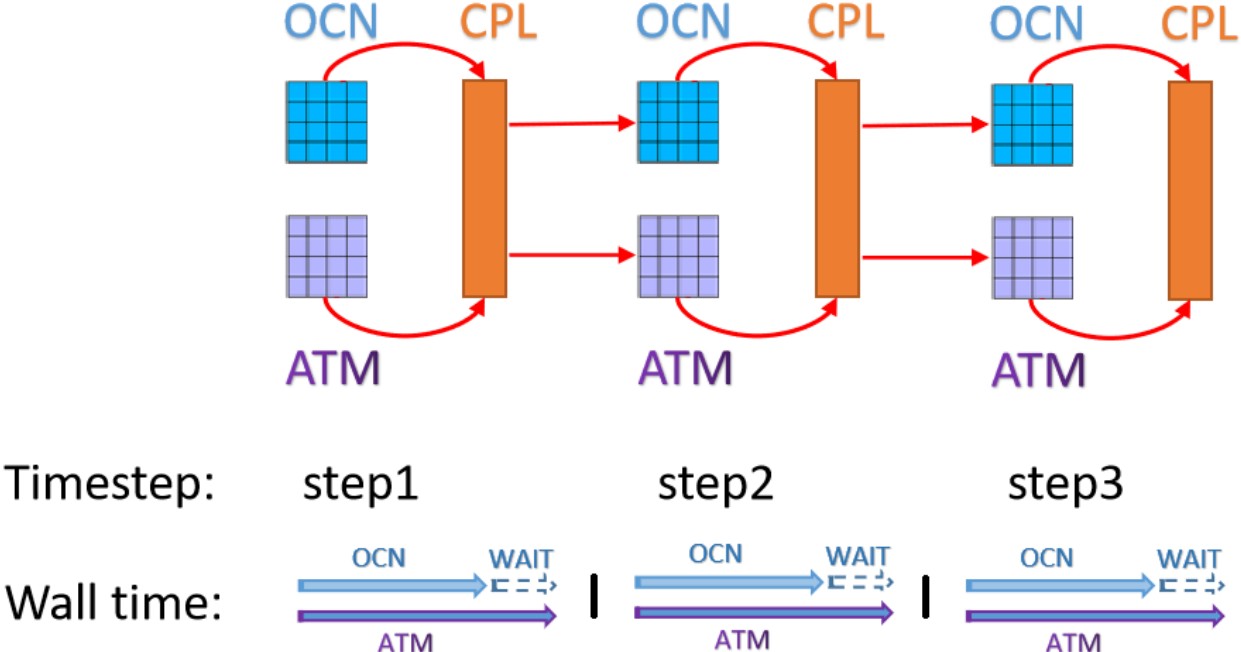

Figure 2: Concurrent mode of the coupled model. The small blocks under OCN or above ATM are the small subdomains in each node; the block under CPL is the coupler. The red curve arrows indicate that the component models are sending data to the coupler and the red straight arrows indicate that the component models are reading data from the coupler. The horizontal arrows in the wall time indicate the time axis of each component model and the ticks on the time axis indicate the coupling time steps.


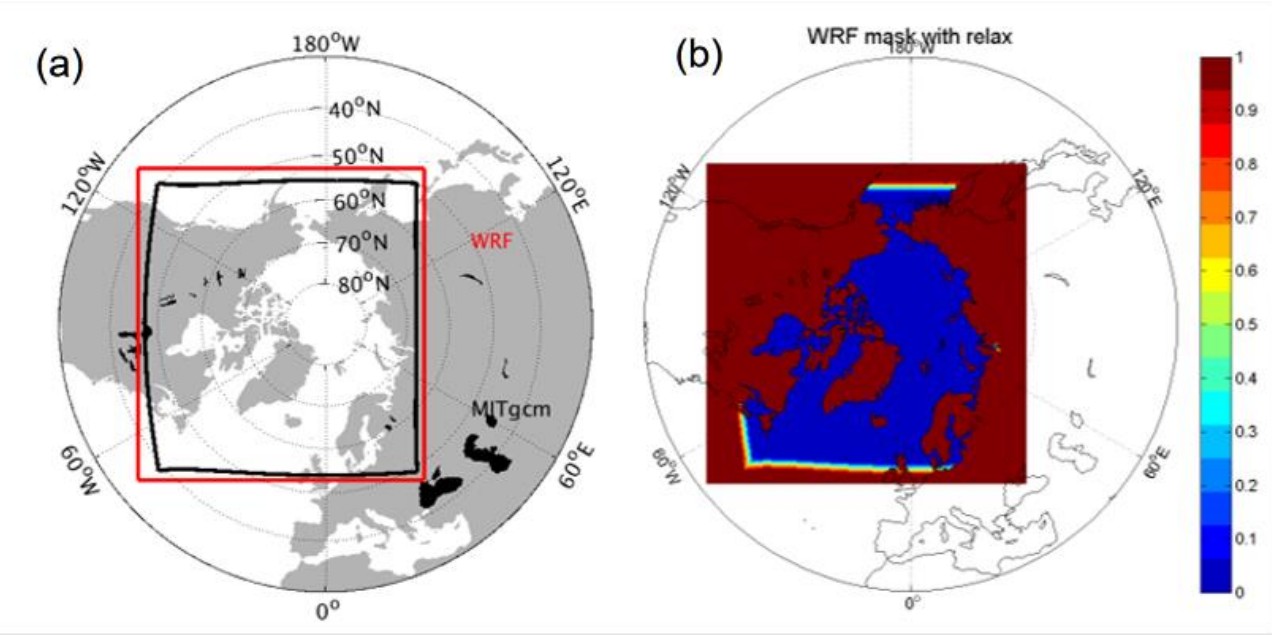


Figure 3: (a) Model domain of the MITgcm and the Polar WRF model. The red and black lines denote the boundaries of the Polar WRF and the MITgcm model, respectively. (b) Relaxation coefficient for the external forcing file of the Polar WRF bottom boundary conditions.

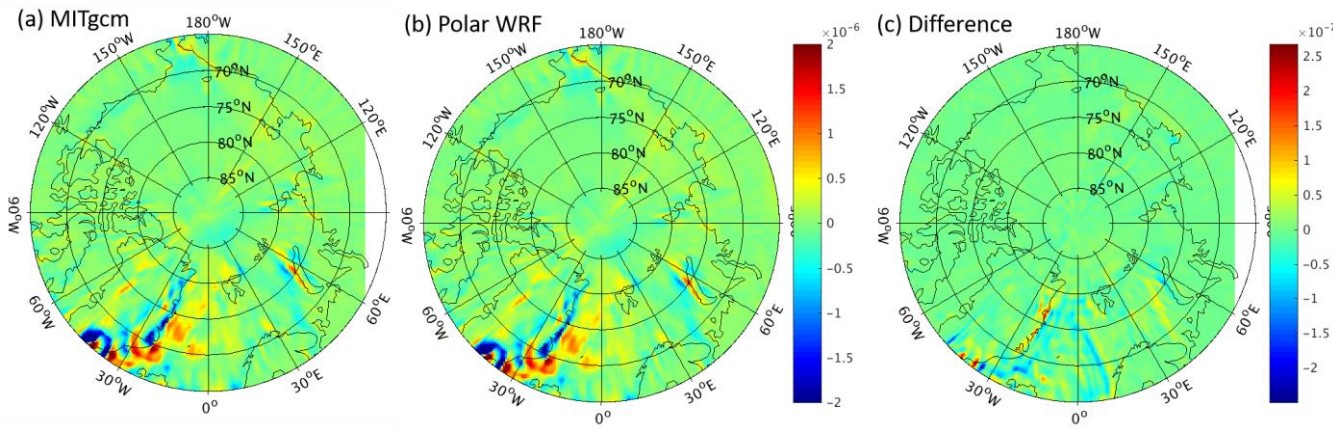


Figure 4: Wind stress curl (unit: Nm-2) derived from (a) the MITgcm output, (b) the Polar WRF output, and (c) their difference on March 1, 2012. The difference of wind stress curl between the Polar WRF and MITgcm is calculated by interpolating the Polar WRF output onto the MITgcm grid.

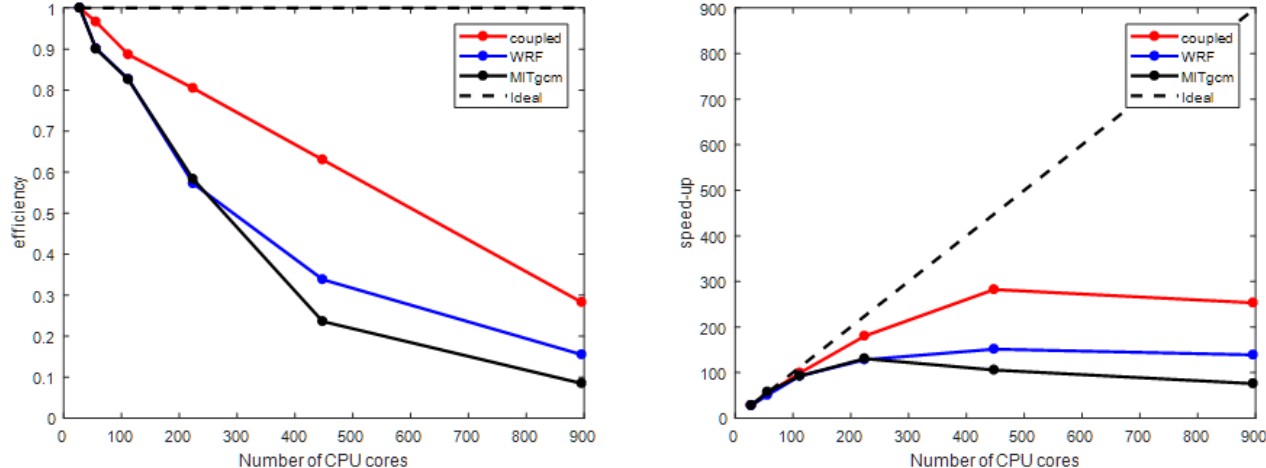


Figure 5: The parallel efficiency (left) and speed-up (right) test of the coupled model and the stand-alone component models, employing up to 896 CPU cores. The simulation using 28 CPU cores is regarded as the baseline case when computing the speed-up. The tests are performed on a Lenovo Blade Server system composed of 240 dual-socket compute nodes based on 14-core Intel Haswell processors.


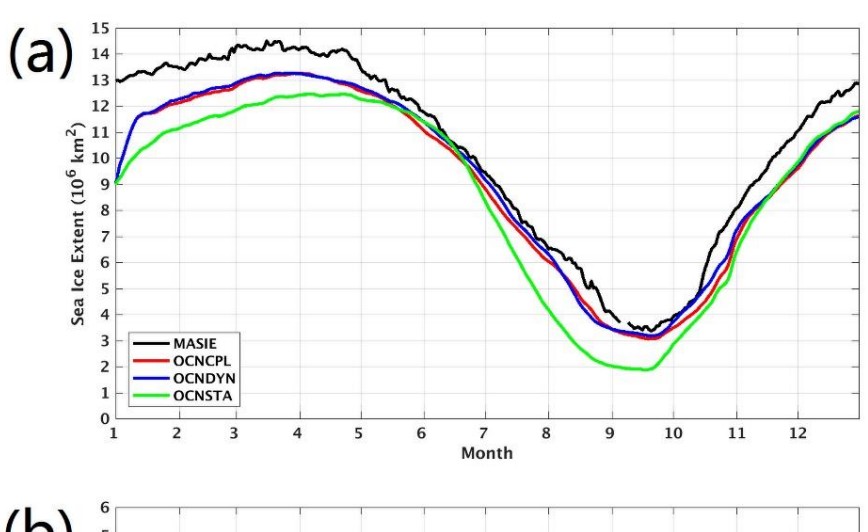

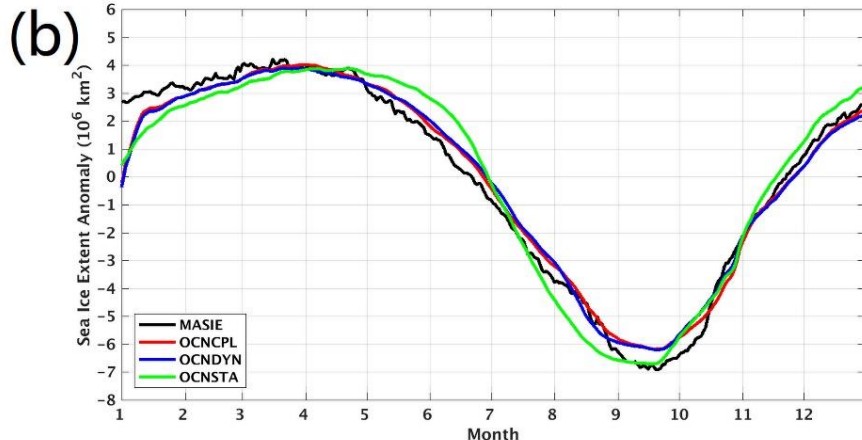

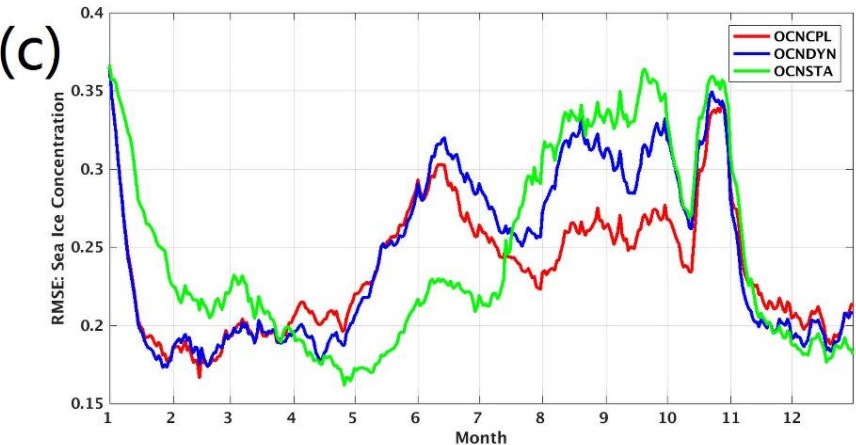

Figure 6: Time series of (a) sea ice extent, (b) sea ice extent anomaly, and (c) root mean square error (RMSE) of modeled sea ice concentration with respect to the OSISAF observation in 2012. The black, red, green and blue lines in (a) denote sea ice extent of the MASIE observation, the OCNCPL run, the OCNSTA run and the OCNDYN run, respectively. The black, red, green and blue lines in (b) denote sea ice extent anomaly of the MASIE observation, the OCNCPL run, the OCNSTA run and the OCNDYN run, respectively. The red, green and blue lines in (c) denote the sea ice concentration RMSE of the OCNCPL run, the OCNSTA run and the OCNDYN run, respectively. MASIE = Multisensor Analyzed Sea Ice Extent; OSISAF = Ocean and Sea Ice Satellite Application Facility.


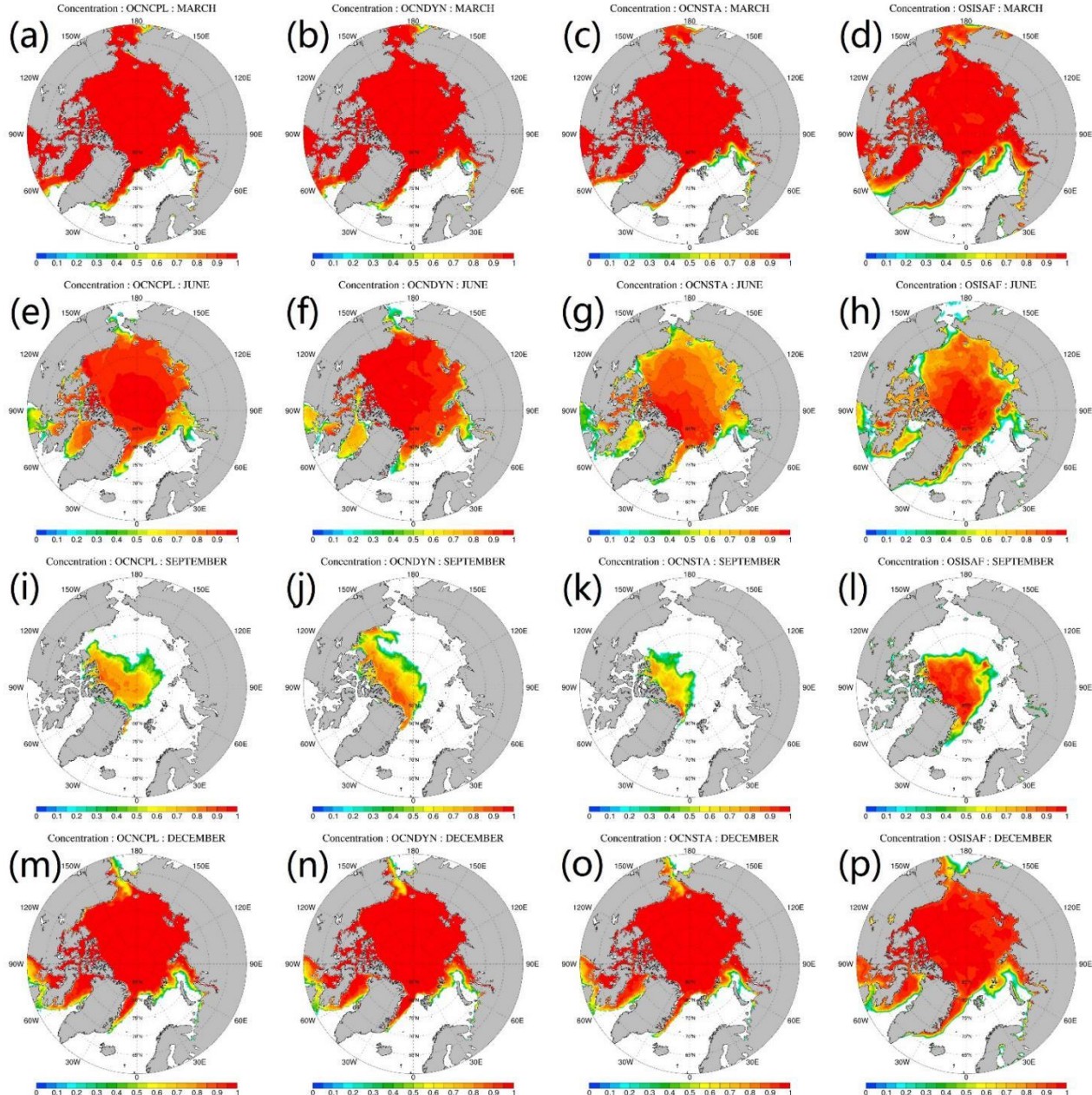

Figure 7: Modeled and observed monthly mean sea ice concentration. From top to bottom panels show the March, June, September and December sea ice concentration, respectively. **From** left **to** right panels show sea ice concentration of the OCNCPL run, the OCNDYN run, the OCNSTA run and the OSISAF observations. OSISAF = Ocean and Sea Ice Satellite Application Facility.

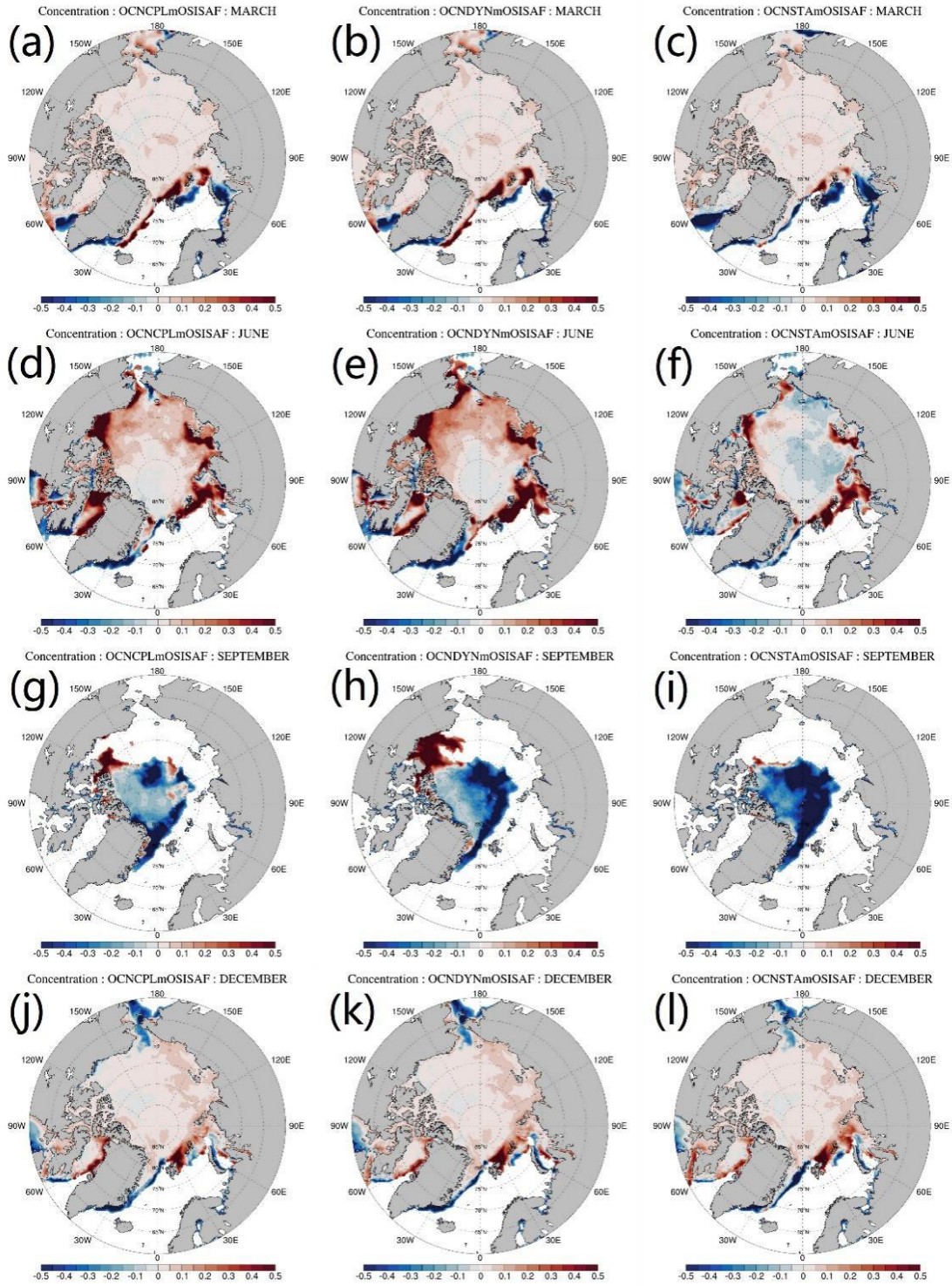

Figure 8: Deviation between the modeled and observed monthly mean sea ice concentration. From top to bottom panels show the March, June, September and December sea ice concentration deviation respect to the OSISAF observations, respectively. The left, middle, and right panels show results of the OCNCPL run, the OCNDYN run, and the OCNSTA run. OSISAF = Ocean and Sea Ice Satellite Application Facility.

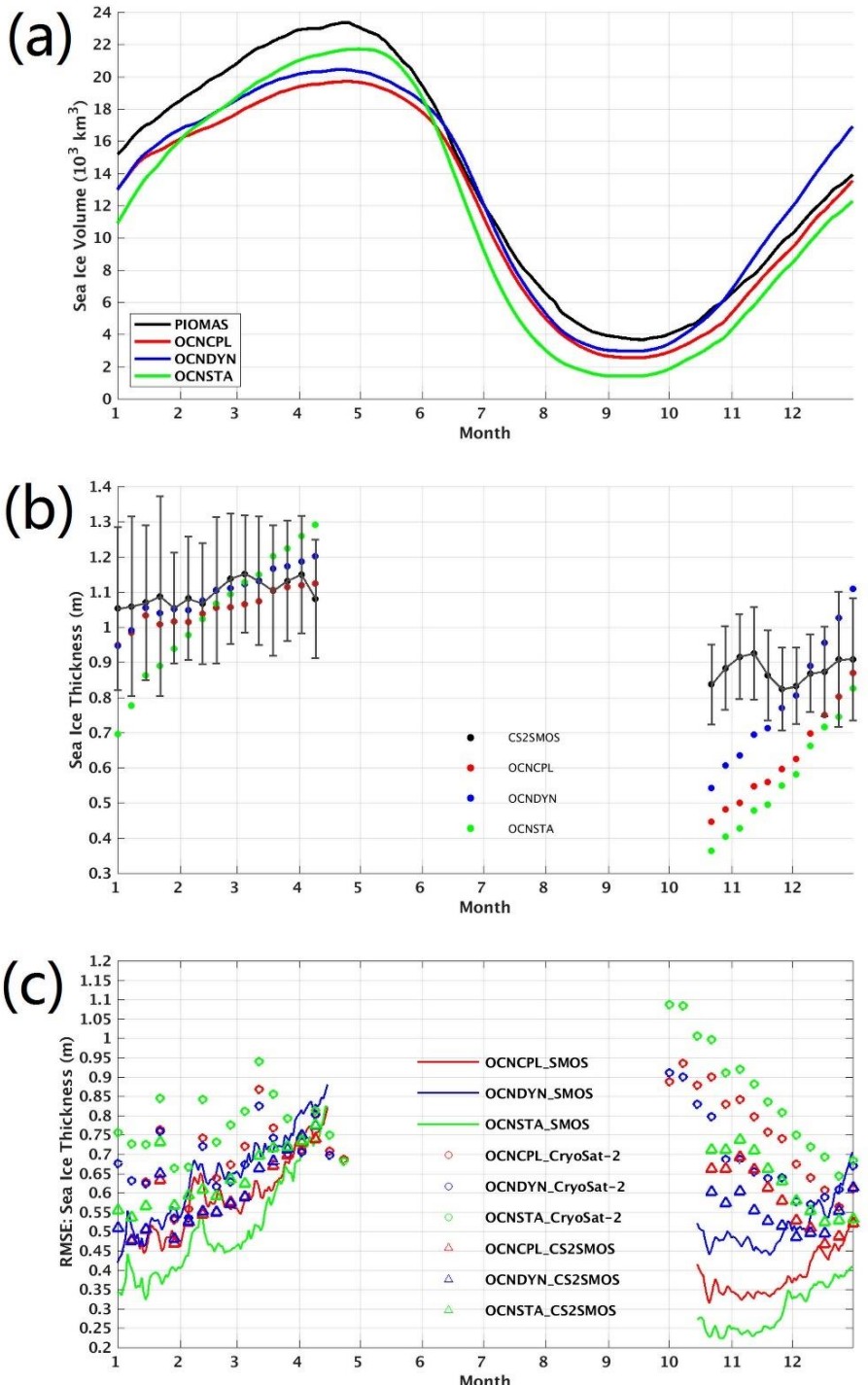

Figure 9: Time series of (a) total sea ice volume, (b) spatial mean sea ice thickness, and (c) the RMSE of sea ice thickness with respect to the satellite-retrieved observations in 2012. The black, red, green and blue lines in (a) denote total sea ice

volume of the PIOMAS data, the OCNCPL run, the OCNSTA run and the OCNDYN run, respectively. The black, red, green and blue dots in (b) denote sea ice thickness of the CS2SMOS observations, the OCNCPL run, the OCNSTA run and the OCNDYN run, respectively. The black bar in (b) represents the observational uncertainties of the CS2SMOS data. The red, green and blue masks in (c) denote sea ice thickness RMSE of the OCNCPL run, the OCNSTA run and the OCNDYN run with respect to the SMOS observations in thin ice (< 1 m) region (line), the Cryosat-2 observations (circle), the CS2SMOS

observations (triangle), respectively. Model grid points without available observations are not taken into the sea ice thickness RMSE calculation. PIOMAS = Pan-Arctic Ice Ocean Modeling and Assimilation System; SMOS = Soil Moisture Ocean Salinity.

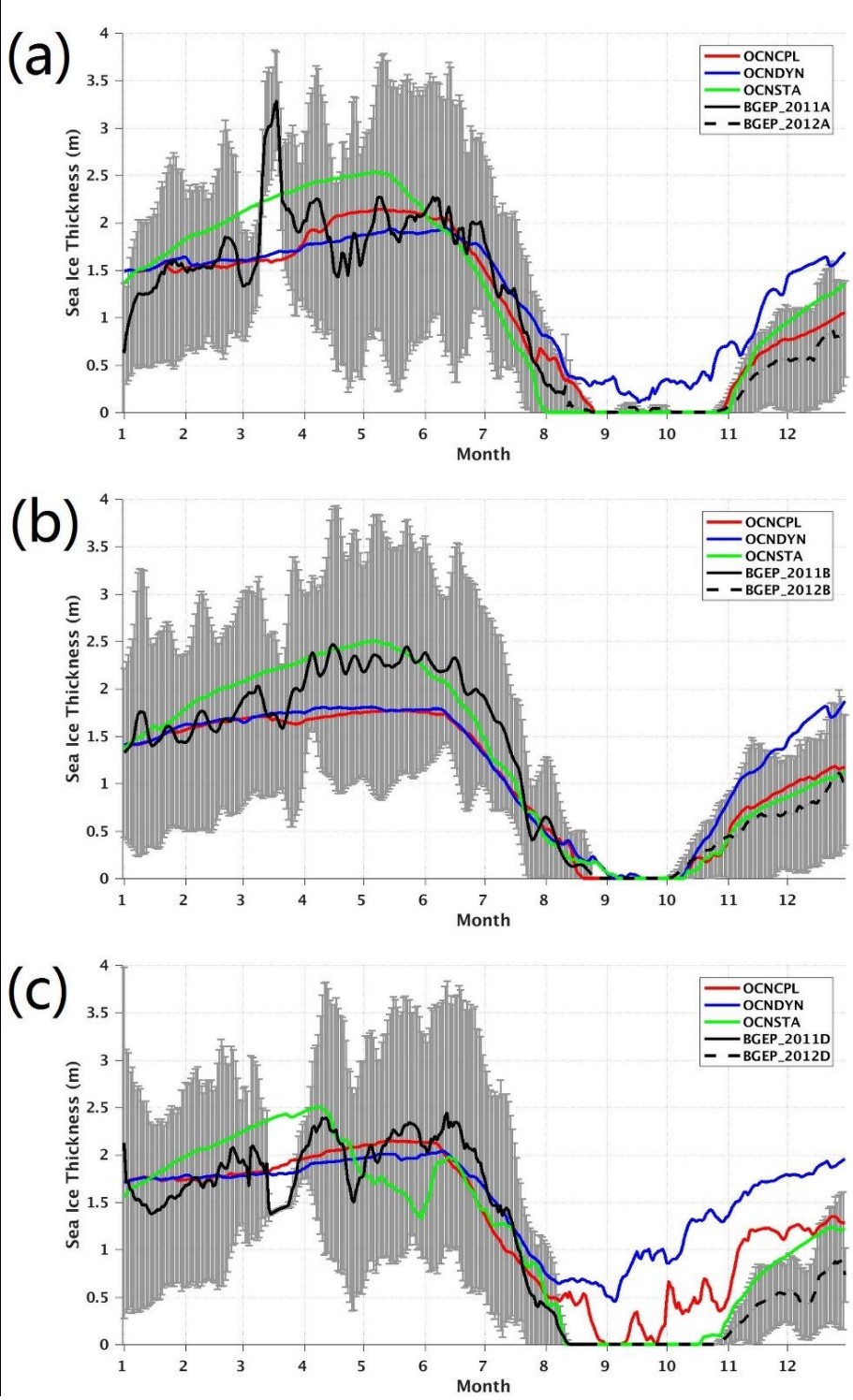

Figure 10: Time series of sea ice thickness at three positions: (a) (75 °N, 150 °W), (b) (78 °N, 150 °W), and (c) (74 °N, 140 °W). The red, blue and green lines denote sea ice thickness of the OCNCPL run, the OCNDYN run and the OCNSTA run, respectively. The black solid and dashed lines denote sea ice thickness observations of the BGEP ULSs, which were deployed in the summers of 2011 and 2012. The black lines of the BGEP ULS observations have been smoothed with the gray bar representing the observational uncertainties. BGEP = Beaufort Gyre Exploration Project; ULS = upward-looking
sonar.

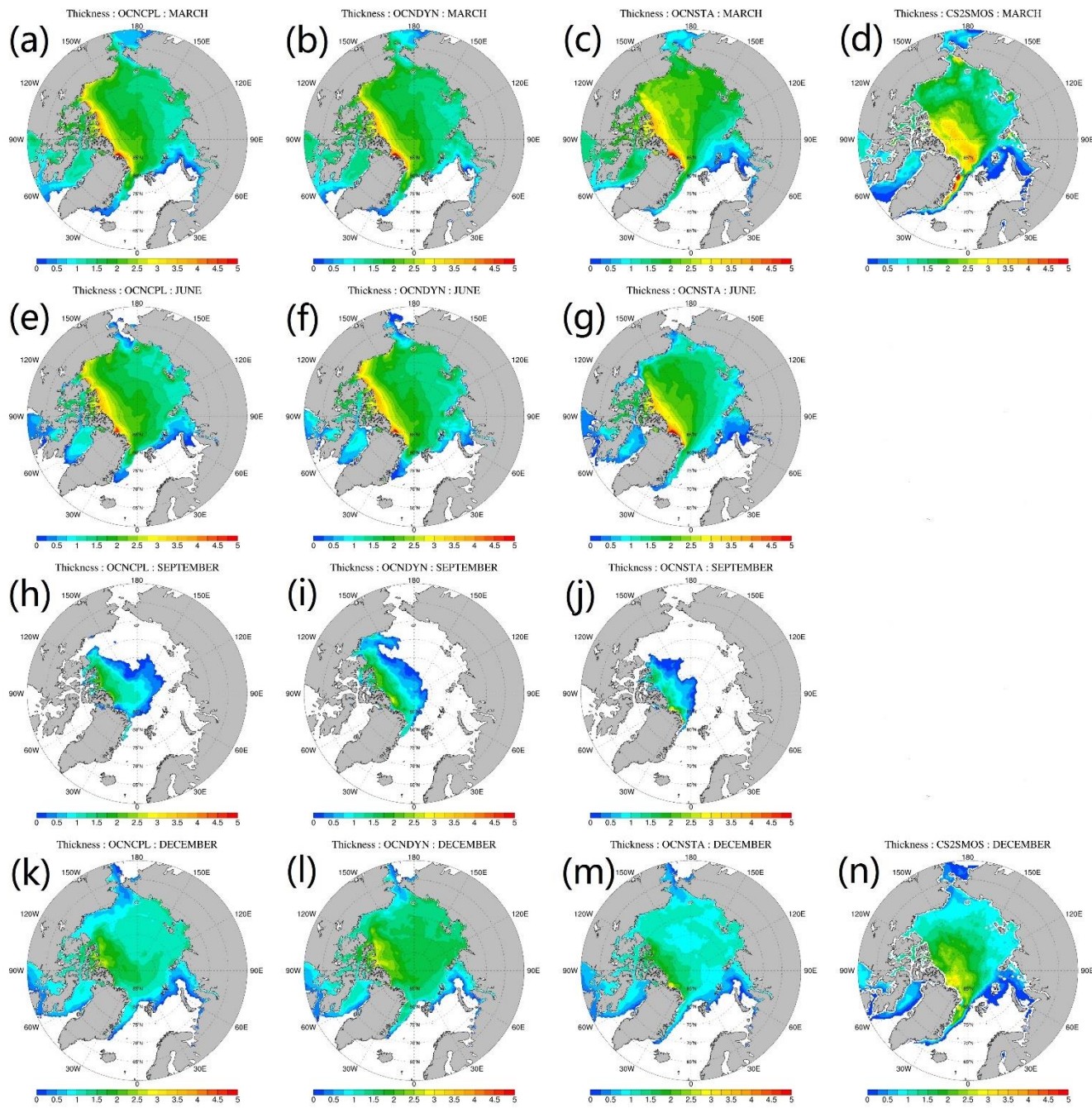

Figure **11**: Monthly mean sea ice thickness. From top to bottom panels show the March, June, September, and December sea ice thickness, respectively. From left to right panels show sea ice thickness of the OCNCPL run, the OCNDYN run, the OCNSTA run and the CS2SMOS data**.**

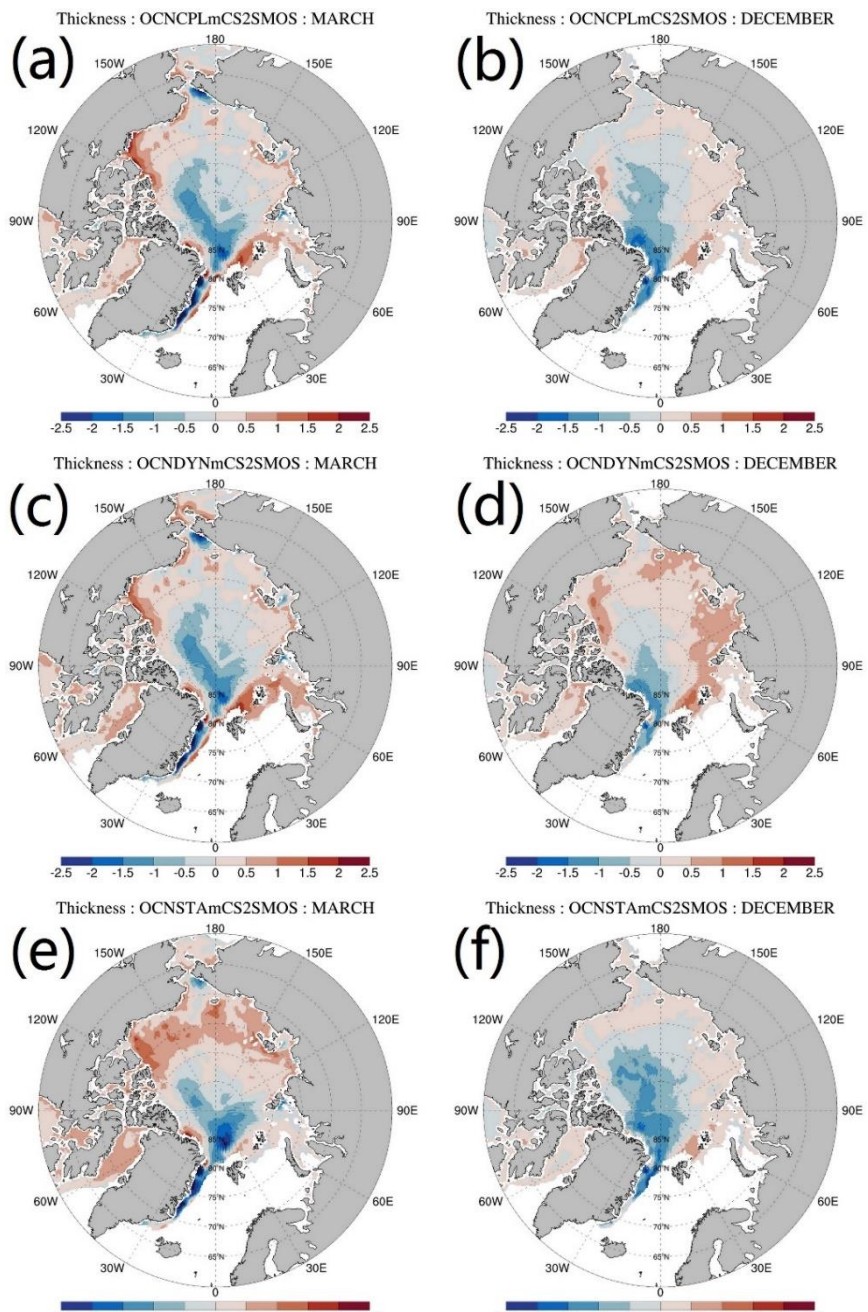

Figure 12: Deviation of the modeled monthly mean sea ice thickness and the CS2SMOS data. The top, middle, and bottom panels show sea ice thickness deviation of the OCNCPL run, the OCNDYN run and the OCNSTA run, respectively. The left and right panels show results in March and December.

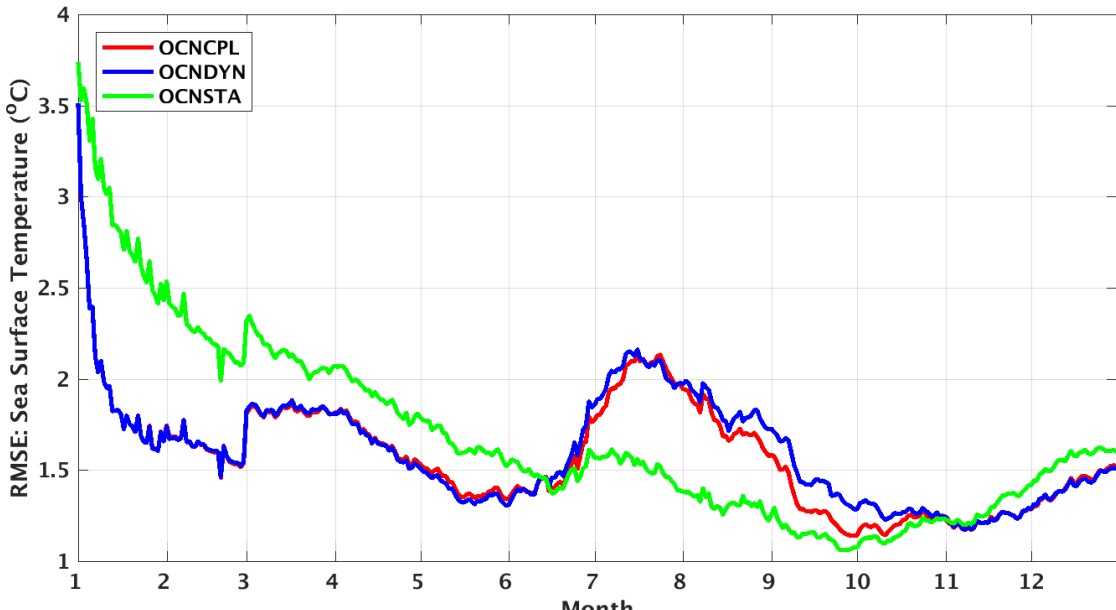

Figure 13: Time series of the RMSE of modeled SST with respect to the GMPE observations in 2012. The red, blue and green lines denote the SST RMSE of the OCNCPL run, the OCNDYN run and the OCNSTA run, respectively. GMPE = Group for High-Resolution Sea Surface Temperature Multi-Product Ensemble.

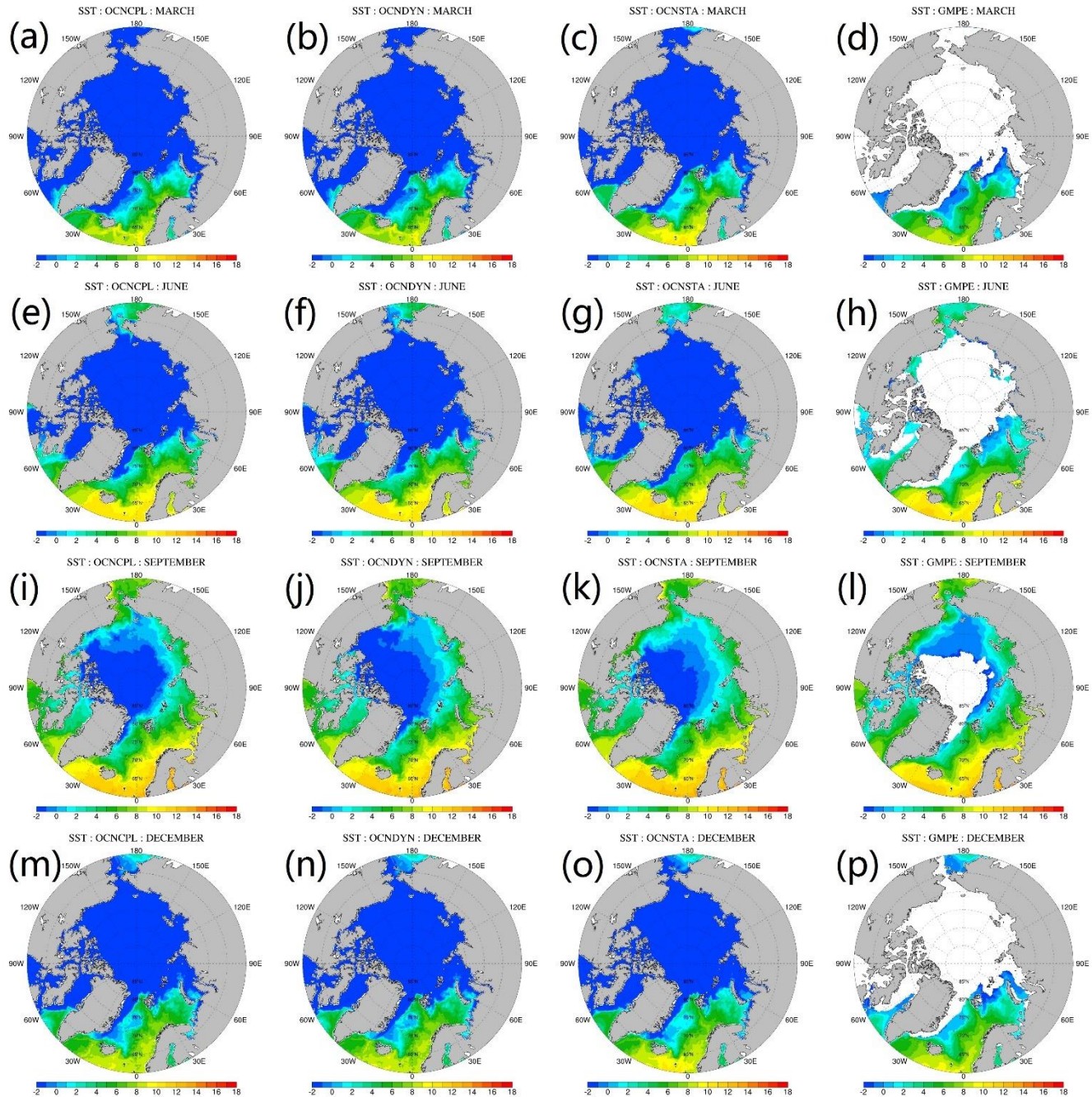

Figure 14: Modeled and observed monthly mean SST. From top to bottom panels show the March, June, September and December SST, respectively. From left to right panels show the SST of the OCNCPL run, the OCNDYN run, the OCNSTA run and the GMPE observations, respectively. GMPE = Group for High-Resolution Sea Surface Temperature Multi-Product Ensemble.

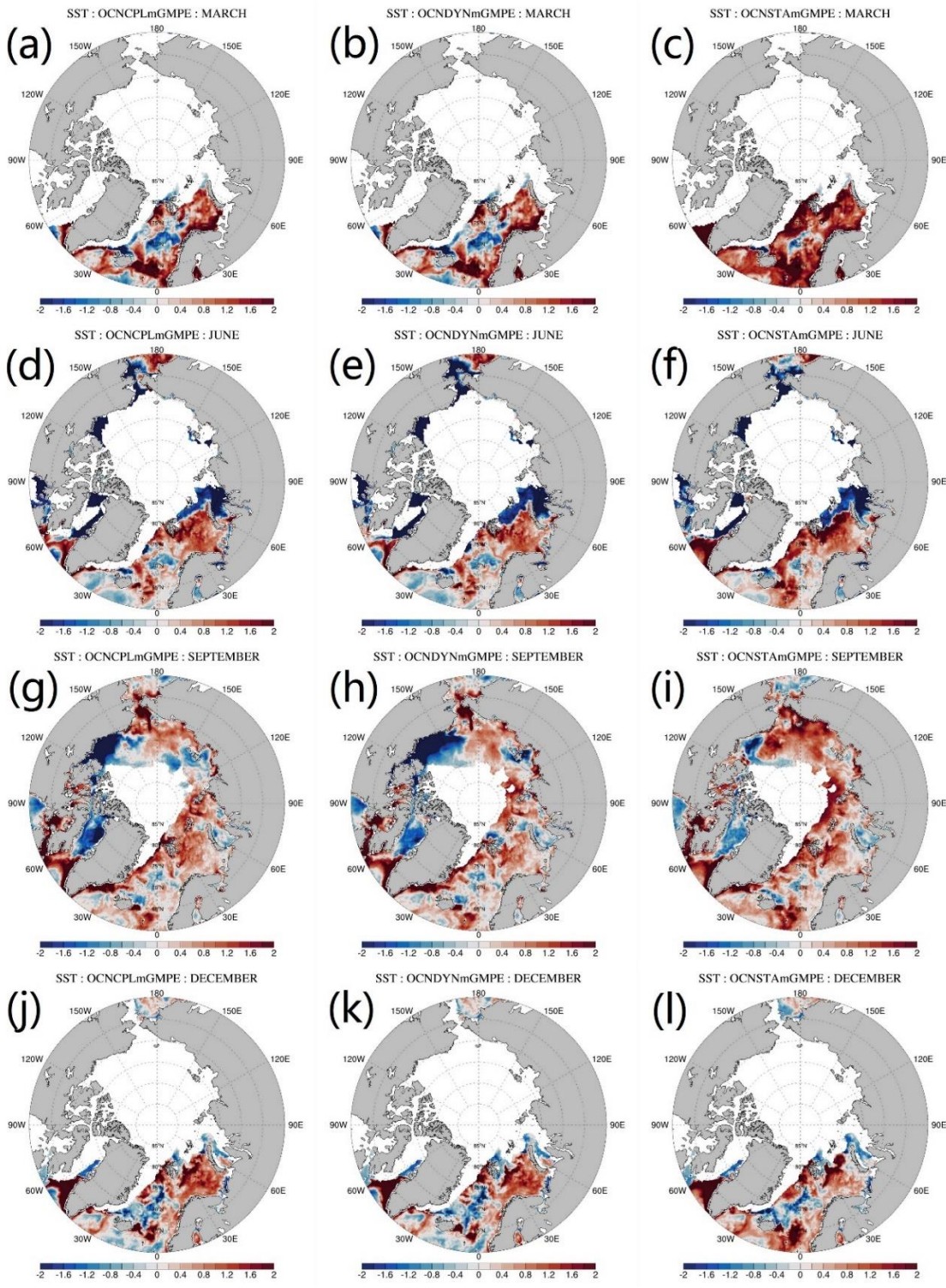

Figure 15: Deviation of the modeled monthly mean SST and the GMPE SST data. From top to bottom panels show the March, June, September and December SST deviation, respectively. From left to right panels show the SST of the OCNCPL run, the OCNDYN run and the OCNSTA run, respectively. GMPE = Group for High-Resolution Sea Surface Temperature Multi-Product Ensemble.


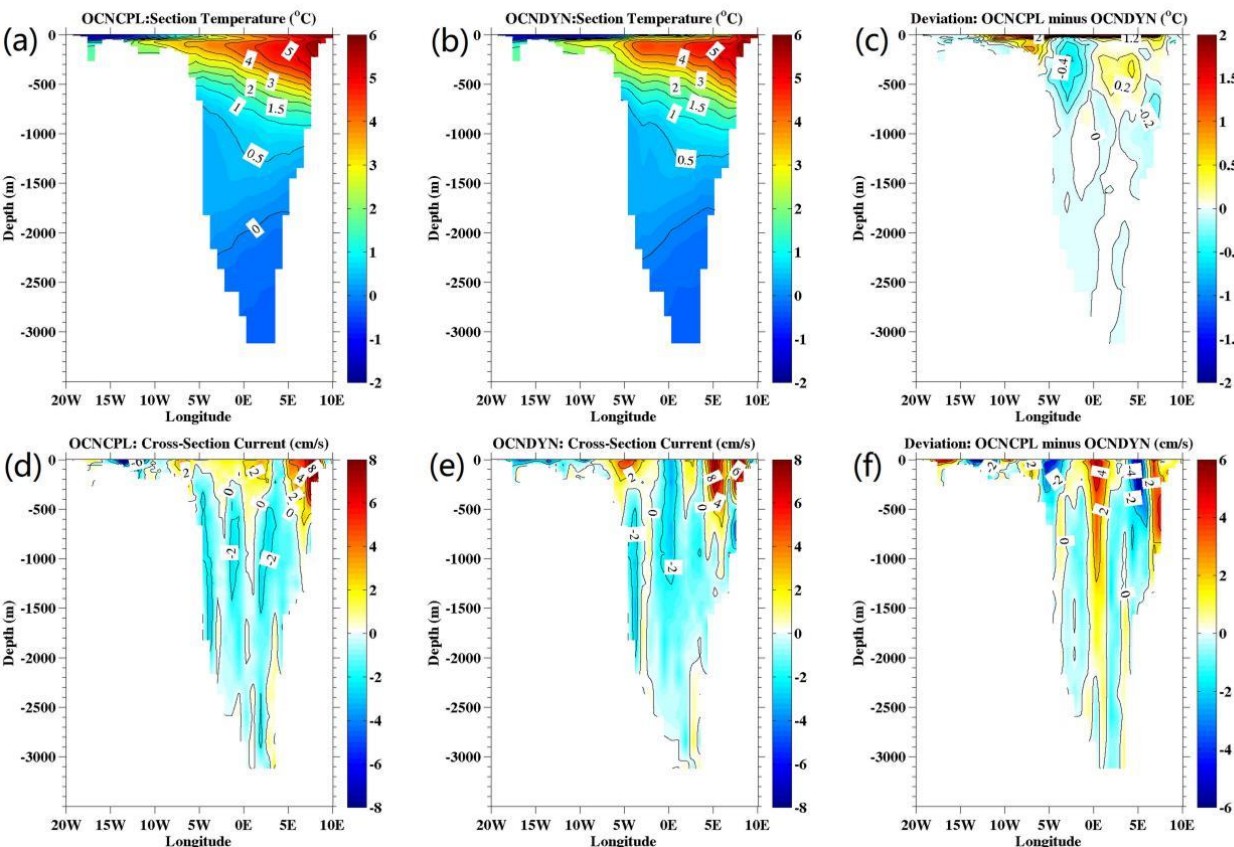

Figure 16: July-August-September mean ocean temperature and meridional velocity section along 78 °N in Fram Strait. The top and bottom panels show the ocean temperature and meridional velocity, respectively. The left, middle, and right panels show the OCNCPL run, the OCNDYN run, and the deviation between them, respectively.

Table 1: The summary of physic option and details of coupled system

| Atmosphere component (Polar WRF) | |
|---|---|
| Horizontal spacing | 27km |
| Horizonal grid points | 306 x 306 |
| Polar WRF time step | 120s |
| Vertical layers | 60 |
| Lateral boundary conditions | CFSR |
| Polar WRF version | 3.7.1 |
| Cumulus parameterization | Grell-Devenyi scheme (Grell and Dévényi, 2002) |
| Microphysics parameterization | WRF single-moment 6-class scheme |
| Longwave and shortwave radiation | Rapid Radiative Transfer Model |
| Boundary layer | Mellor-Yamada-Janjic scheme |
| Land surface | Unified Noah LSM (Chen & Dudhia, 2001) |
| Ocean/sea ice component (MITgcm) | |
| Horizontal spacing | 18km |
| Horizonal grid points | 420 x 384 |
| MITgcm time step | 1200s |
| Vertical layers | 50 |
| Lateral boundary conditions | ECCO2 |
| MITgcm version | checkpoint64a |
| Equation of sea water state | Jackett and McDougall (1995) |
| Vertical mixing scheme | K-profile parameterization (KPP) scheme |

| | |
|---|---|
| Horizontal advection scheme | Seventh-order monotonicity-preserving advection scheme (Daru and Tenaud, 2004) |
| Ice rheology | Viscous-Plastic constitutive law |
| Ice momentum solver | Line successive over-relaxation (Zhang and Hibler, 1997) |
| Ice thermodynamics | Zero-layer snow/ice thermodynamics (Semtner, 1976 ) |
| Albedo (under CFSR forcing) | dry ice: 0.65  wet ice: 0.55  dry snow: 0.8  wet snow: 0.7 |
| **Coupler component (C-Coupler)** | |
| Coupler version | C-Coupler2 |
| Coupling frequency | 1200s |
| Interpolation scheme | Bilinear remapping algorithm |
| Coupling parameters (from MITgcm to Polar WRF) | SST, sea ice concentration, sea ice thickness, snow depth, ice surface albedo |
| Coupling parameters (from Polar WRF to MITgcm) | Downward longwave radiation, downward shortwave radiation, 10 m wind speed, 2 m air temperature, 2 m air specific humidity, precipitation |


Table 2: Comparison of CPU time spent on coupled and stand-alone runs. The CPU time spent on two stand-alone simulations are presented to show the difference between coupled and stand-alone simulations. 'total_cpu_number' denotes the requested CPUs, 'total_run_time' denotes the total CPU elapsed time. 'wrf_interface', 'wrf_integration', 'mitgcm_interface' and 'mitgcm_integration' denote the CPU elapsed time used for coupling interface by the WRF, numerical integration by the WRF, coupling interface by the MITgcm, and numerical integration by the MITgcm, respectively. 'wrf_time_alone' denotes the CPU elapsed time of the stand-alone WRF runs. 'mitgcm_time_alone' denotes the CPU elapsed time of the stand-alone MITgcm runs. Each run is integrated for 7 model days.

| total_cpu_number | cpu_number_on_each_component_model | total_run_time (unit: s) | wrf_interface (unit: s) | mitgcm_interface (unit: s) | wrf_integration (unit: s) | wrf_time_alone (unit: s) | mitgcm_integration (unit: s) | mitgcm_time_alone (unit: s) |
|---|---|---|---|---|---|---|---|---|
| 28 | 14 | 12840 | 4.8 | 12131 | 12835.2 | / | 709 | / |
| 56 | 28 | 12000 | 4.74 | 11196 | 11995.26 | 7140 | 804 | 317 |
| 112 | 56 | 10440 | 5.16 | 6477 | 10434.84 | 3960 | 3963 | 154 |
| 224 | 112 | 3780 | 5.26 | 3550 | 3774.74 | 2160 | 230 | 96 |
| 448 | 224 | 2460 | 5.21 | 2116 | 2454.79 | 1560 | 344 | 68 |
| 896 | 448 | 1380 | 358 | 48 | 1022 | 1320 | 1332 | 84 |


Table 3: The initial conditions, boundary conditions and forcing terms used in the experiments.

| Experiments description | | | |
|---|---|---|---|
| Experiment name | Description | Bottom boundary forcing for atmospheric component | Surface boundary forcing for ice/oceanic component |
| OCNCPL | two-way coupled simulation | MITgcm | Polar WRF |
| OCNDYN | one-way coupled simulation that the MITgcm only receives the variables from the Polar WRF, but without sending variables back to the Polar WRF | CFSR | Polar WRF |
| OCNSTA | stand-alone MITgcm simulation | Not used | CFSR |

Note:

Atmospheric initial and boundary conditions: CFSR

Oceanic boundary conditions: ECCO2

Oceanic initial conditions: restart field on 1 January 2012 derived from a stand-alone MITgcm simulation initialized from climatological temperature and salinity field derived from WOA05 and forced by the 3-hourly JRA55 data from 1979 to 2011

Table 4: Monthly mean northward cross-section velocity (cm/s) and temperature (°C) averaged between 5°E and 8°40'E at 78°50'N in Fram Strait. A1 represents algorithm 1 that values are calculated from sea water with potential temperature higher than 1°C. A2 represents algorithm 2 that values are calculated from sea water with potential temperature higher than -0.1°C. A3 represents algorithm 3 that values are calculated from sea water with depth shallower than 700 m. The observations are averaged between 1998 and 2003. WSCOBS = West Spitsbergen Current Observation.

|  |  | July | | August | | September | |
|---|---|---|---|---|---|---|---|
|  |  | $V_{mean}$ | $T_{mean}$ | $V_{mean}$ | $T_{mean}$ | $V_{mean}$ | $T_{mean}$ |
| A1: (T>1°C) | OCNCPL | 3.94 | 3.56 | 4.03 | 3.66 | 4.03 | 4.02 |
|  | OCNDYN | 3.22 | 3.69 | 2.93 | 3.79 | 2.27 | 3.91 |
|  | WSCOBS | 6.26 | 2.76 | 6.98 | 2.90 | 7.36 | 3.02 |
| A2: (T>-0.1°C) | OCNCPL | 3.53 | 2.30 | 3.32 | 2.35 | 3.24 | 2.54 |
|  | OCNDYN | 2.63 | 2.58 | 2.38 | 2.69 | 1.98 | 2.66 |
|  | WSCOBS | 5.82 | 2.35 | 6.39 | 2.44 | 6.69 | 2.51 |
| A3: (0-700 m) | OCNCPL | 4.21 | 3.97 | 4.33 | 4.03 | 4.16 | 4.53 |
|  | OCNDYN | 3.87 | 4.36 | 3.53 | 4.54 | 2.55 | 4.65 |
|  | WSCOBS | 6.09 | 2.61 | 6.67 | 2.72 | 7.04 | 2.83 |