# Peer review of "A fully coupled Arctic sea ice-ocean-atmosphere model (ArcIOAM v1.0) based on C-Coupler2: model description and preliminary results"

_Geoscientific Model Development, 2020_

## Referee Comment (RC1) · Anonymous Referee #1 · 1 Aug 2020

This manuscript describes a coupled via C-Coupler2 Arctic ocean sea ice configuration of MITgcm and Polar WRF atmosphere. The model is intended for high quality Arctic sea ice seasonal predictions. There is large demand for high quality regional climate models of the Arctic basin and such activity must be strongly appreciated. In the manuscript the setup has been validated for year 2012 because of a strong storm formed off the coast of Alaska on 5 August 2012. The role of sea ice-ocean-atmosphere interaction has been addressed. Although the authors demonstrate good modelling skills, good knowledge of the Arctic Ocean system and impressive level of model validation the paper in it's present form failed to convince me that the new climate model is ready to use and that is better than any existing global climate model.

My main criticism is that (1) the performance of the model on different HPC systems, regarding scalability and the costs of all individual components is not addressed, (2) the presented configuration is not properly tuned. I encourage this paper for resubmission after these weaknesses have been fixed.

In the model description, there need to be discussion on why fields and not fluxes are coupled. Do the authors guarantee the same bulk formulas are employed on the atmospheric and ocean sides? Which difference is expected if the fluxes are coupled? I guess that COSMO-CLM/NEMO group has some experience with it although not with the Arctic region. I find this aspect is more important than describing the computation of the corner geographic information for MITgcm. The latter piece I would even omit due to its simplicity. A following chapter after the model description, which gives more information about the model scalability and cost is required.

I believe that C-Coupler is a good tool to use but the statement that a model produces bitwise identical results with a different coupler means that the coupler just works. Is it better in terms of performance? Which interpolation option do you use (question is more relevant for the wind stress)? As an illustration it would be good to see the curl of the wind stress on the atmospheric and oceanic meshes (instead of Fig.3).

In the OCNDYN run is it just setting alpha to 1.0? I believe that one year of coupled simulation is too short to validate the model. Either ensemble of runs or a longer simulation is desired here. I don't want to force the authors to do much of additional work but the cheapest way would be to add the stand-alone MITgcm run for the comparison here. The spin up with JRA55 has been already computed.

Section 4.1, which introduces Fig. 5 illustrates that the model has not been tuned properly yet. Although the authors (line 254) claim that OCECPL is closer to observations, which might be true, but I see that both runs failed. Here it is important to give (at least visually) the measure of the error. A stand-alone MITgcm run, hence, would be a good choice. Provided the high skill of validation made in the following chapters I assume

that the model has to be better tuned first.

Minor things: line 181: "...without any data assimilation...". There was nothing said about new model is within an assimilation framework before or after. Why to mention this? line 182: "...the coupled model free simulations..." what do you mean? line 188: indeed, nothing about atmosphere. Patterns of SLP although would be of interest. line 230: too speculative. did you do another run with different albedo? line 290: I would elaborate more on this if possible. line 366: not really or show that it is better than in other (global) models line 372: is it due to the albedo feedback? In OCECPL it is computed on the MITgcm side. What happens in OCEDYN? Again, figures 6c, 7 would have more value if the model is tuned. In the present form the model is not ready for this validation.

---

## Referee Comment (RC2) · David Bailey (Referee) · 18 Sep 2020

This manuscript describes a "new" regional coupled model for the Arctic intended for short-term forecasting and shows some preliminary simulation results. My main issue with this manuscript is that I do not see the novelty here. Similar regional models such as ARCSyM (Lynch et al. 1995), RASM (Cassano, Maslowski, others), Rinke et al. 2000 have not been mentioned here. The "novel" aspect seems only to be the coupling of the MITgcm to the Polar WRF model and perhaps the new C-Coupler. I nearly recommended reject for this manuscript. It seems more like a technical report. Also, the English usage here is confusing and there are a number of grammatical errors

that I do not have time to go into here. Here are some specific suggestions that might make this worthy of publication in GMD.

1. There needs to be more on the novel aspects here. What does this model provide specifically that previous models did not? What time scales is this intended for? Short-term forecasting of weeks? Seasons?

2. There is no mention of the land component. Is this just the imbedded Polar WRF component? Why is this not important to the Arctic simulations?

3. The MITgcm sea ice component is quite old and simplistic. What is the albedo formulation on sea ice? What about on land? What about a sea ice thickness distribution?

4. The authors suggest that CMIP5 models represent sea ice in a simple way, but MITgcm is not any more complicated. A number of CMIP5 models used the Los Alamos Sea Ice Model (CICE) which is leading edge. What about CMIP6?

5. The grid staggering discussion and Figure 3 are really not necessary.

6. The case study they use is the year 2012. The boundary forcing for the atmosphere is from NCEP/NCAR re-analysis and for the ocean from the ECCO model. The sea ice extent is reasonable, but the sea ice volume is biased low compared to PIOMAS. I believe the problem here is the simplicity of the sea ice thermodynamics and lack of a subgridscale thickness distribution. I feel like the model might be tuned for 2012 and want to see how it performs for other years. Some comparison to IceSAT data would also be beneficial here.

---

## Author Comment (AC2) · 16 Oct 2020

**Reply to Reviewer Comment 2 (RC2)**

*Comment:*

*This manuscript describes a "new" regional coupled model for the Arctic intended for short-term forecasting and shows some preliminary simulation results. My main issue with this manuscript is that I do not see the novelty here. Similar regional models such as ARCSyM (Lynch et al. 1995), RASM (Cassano, Maslowski, others), Rinke et al. 2000 have not been mentioned here. The "novel" aspect seems only to be the coupling of the MITgcm to the Polar WRF model and perhaps the new C-Coupler. I nearly recommended reject for this manuscript. It seems more like a technical report. Also, the English usage here is confusing and there are a number of grammatical errors that I do not have time to go into here. Here are some specific suggestions that might make this worthy of publication in GMD.*

Reply:

The authors thank the reviewer for the insightful comments. Thanks for pointing out that we omit similar works about regional coupled models. We have added these work as reference and re-written the introduction. The motivation of our work is to couple the Arctic ocean-seaice configuration of the MITgcm which is operationally running in our institute (ArcIOPS; Liang et al., 2020), to the Arctic atmospheric model (Polar WRF) in order to get a reasonable seasonal sea ice simulation. Beyond the scope of this work, our destination is operational sea ice seasonal prediction based on the coupled model and data assimilation algorithm. In my opinion, the novelty of our study is that we couple the Polar WRF and MITgcm which both featured with good performances in polar regions for the first time in the Arctic region and that we have solved some technical issues during the coupling process with a new coupler.

We have revised our manuscript by adding discussion on sea ice dynamic during coupling process. We have better motivated the manuscript in the introduction. We have proof-read the manuscript carefully to make it more readable.

*Comment :*

*1. There needs to be more on the novel aspects here. What does this model provide specifically that previous models did not? What time scales is this intended for? Short-term forecasting of weeks? Seasons?*

Reply:

Sun et al. (2019) introduced a regional ocean-atmosphere coupled model covering the Red Sea based on the MITgcm and WRF. The novelty of our study is that we couple the Polar WRF and the MITgcm for the first time in the Arctic region and that we have solved some technical issues during the coupling process with a new coupler.

Both Polar WRF and MITgcm have specific features designed for polar regions, we speculate that coupling them will help us to improve seasonal sea ice prediction. We have used MITgcm model to make an operational Arctic synoptic-scale sea ice forecast for several years and showed reasonable results in aspects of sea-ice forecast in synoptic scale (ArcIOPS; Mu et al., 2019; Liang et al., 2020). This work is motivated by the need of a coupled Arctic sea ice-ocean-atmosphere model system for operational sea ice prediction of seasonal timescale (such as initialized at June and predict September sea ice).

*Comment:*

*2. There is no mention of the land component. Is this just the imbedded Polar WRF component? Why is this not important to the Arctic simulations?*

Reply:

We apologize that we overlooked the description of land component in the section of Polar WRF. Yes, land component is embedded inside the Polar WRF. This Polar WRF incorporates many modifications to the standard version of the WRF. These adjustments are described by Bromwich et al. (2009), for example, adjustments to the surface parameterizations. The changes made in the Noah land surface model (LSM;

Chen and Dudhia 2001) include using the latent heat of sublimation for calculating latent heat fluxes over ice surfaces, increasing the snow albedo and the emissivity value for snow, adjusting snow density, modifying thermal diffusivity and snow heat capacity for the subsurface layer, changing the calculation of skin temperature, and assuming ice saturation in calculating the surface saturation mixing ratio over ice. Land component is absolutely important to the Arctic simulation, however at current stage, our coupled model has not capacity of coupling an individual land model, instead, we use the embedded land component in the Polar WRF for technical simplicity. We have added the introduction of land component in the section of Polar WRF description.

*Comment:*

*3. The MITgcm sea ice component is quite old and simplistic. What is the albedo formulation on sea ice? What about on land? What about a sea ice thickness distribution?*

Reply:

There are two calculations of surface albedo provided in the MITgcm.

1)  from LANL CICE model:

$$\alpha = f_s \alpha_s + (1 - f_s)(\alpha_{i_{min}} + (\alpha_{i_{max}} - \alpha_{i_{min}})(1 - e^{-h_i/h_\alpha}))$$

where $f_s$ is 1 if there is snow, 0 if not; the snow albedo, $\alpha_s$ has two values depending on whether $T_s < 0$ or not; $\alpha_{i\min}$ and $\alpha_{i\max}$ are ice albedo for thin melting ice, and thick bare ice respectively, $h_i$ is snow depth, and $h_\alpha$ is a scale height.

2)  From GISS model (Hansen et al 1983):

$$\alpha = \alpha_i e^{-h_s/h_a} + \alpha_s(1 - e^{-h_s/h_a})$$

where $\alpha_i$ is a constant albedo for bare ice, $h_s$ is snow depth, $h_a$ is a scale height

and $\alpha_s$ is a variable snow albedo

$$\alpha_s = \alpha_1 + \alpha_2 e^{-\lambda_a a_s}$$

where $\alpha_1$ is a constant, $\alpha_2$ depends on $T_s$, $\alpha_s$ is the snow age, and $\lambda_a$ is a scale frequency.

In our coupled model, surface albedo from LANL CICE model is used. Land component is from atmospheric component of Polar WRF mentioned in last reply.

In order to parameterize a sub-grid scale distribution for sea ice thickness, the mean sea ice thickness in each grid can be split into as many as 7 thickness categories in the MITgcm sea ice model. In our coupled model for simplicity, we use 2 thickness categories: open water and sea ice.

*Comment:*

*4. The authors suggest that CMIP5 models represent sea ice in a simple way, but MITgcm is not any more complicated. A number of CMIP5 models used the Los Alamos Sea Ice Model (CICE) which is leading edge. What about CMIP6?*

Reply:

We apologize for the improper statement about sea ice representation in the CMIP5 models in the original manuscript. The sea ice physical mechanism is rather complicate in the CICE. We have replaced the improper words in the manuscript. We have also gone through the entire manuscript to revise other improper words.

*Comment:*

*5. The grid staggering discussion and Figure 3 are really not necessary.*

Reply:

We agree with the reviewer that the grid staggering discussion and Figure 3 are not necessary. So the part of computation of corner information and Fig. 3 are removed from the revised manuscript.

*Comment:*

*6. The case study they use is the year 2012. The boundary forcing for the atmosphere is from NCEP/NCAR re-analysis and for the ocean from the ECCO model. The sea ice extent is reasonable, but the sea ice volume is biased low compared to PIOMAS. I believe the problem here is the simplicity of the sea ice thermodynamics and lack of a subgridscale thickness distribution. I feel like the model might be tuned for 2012 and want to see how it performs for other years. Some comparison to IceSAT data would also be beneficial here.*

Reply:

We agree that one year is short to validate the model. So we add the runcase of 2016 and add the standalone MITgcm for the comparison. To keep consistency between the coupled model and standalone MITgcm model, the standalone MITgcm simulation is forced by surface variables derived from the CFSR data and uses the same albedo parameters to the coupled model. As the IceSat data covers 2003-2008, we can not validate the sea ice thickness simulations of 2012 and 2016 with the IceSat data. Instead, we choose Cryosat2 and SMOS data to validate sea ice thickness.

In our original manuscript, we make a mistake when calculate sea ice extent in Figure 4a and 4b, which leads to the negative sea ice extent bias in Figure 4a. We confuse sea ice area with sea ice extent. Actually the blue and red curves in Figure 4a and 4b in the original manuscript are sea ice area. We redraw the modeled sea ice extent and add the standalone MITgcm run for the comparison (see the following figure). Compared with the standalone MITgcm run, the modeled sea ice extent in the coupled runs are more closer to the observation. With respect to the one-way coupled run, the spatial distribution of summertime sea ice concentration in the two-way coupled run is more closer to the OSISAF observation.

[Figure]

Fig.1 Time series of (a) sea ice extent, (b) sea ice extent anomaly, and (c) root mean square error (RMSE) of modeled sea ice concentration with respect to the OSISAF observation in 2012. The black, red, blue and green lines in (a) denote sea ice extent of the MASIE observation, the OCNCPL run, the OCNDYN run, and the OCNSTA run respectively. The black, red, blue and green lines in (b) denote sea ice extent anomaly of the MASIE observation, the OCNCPL run, the OCNDYN run, and the OCNSTA run respectively. The red, blue and green lines in (c) denote the sea ice concentration RMSE of the OCNCPL run, the OCNDYN run, and the OCNSTA run, respectively.

Results of the modeled sea ice in 2016 are shown as follows:

The year of 2016 is selected because of the next anomalous sea ice extent minima event after 2012 appeared in 2016. We also conduct two-way coupled run, one-way coupled run, standalone MITgcm run for the year of 2016. The initial fields on 2016.1.1 of the MITgcm and the WRF model are derived from the standalone MITgcm simulation forced by JRA55 data and from the CFSR reanalysis data. Time evolution of modeled sea ice extent shows that the two-way coupled run produces more reasonable sea ice extent compared with the MASIE data, although the modeled September sea ice concentration of the two-way run in the Arctic Pacific section overmelts.

[Figure]

Fig.2 Same to the above figure but for the year of 2016.

[Figure]

Fig.3 Monthly mean sea ice concentration in 2016. The 1$^{st}$, 2$^{nd}$, 3$^{rd}$, 4$^{th}$ row denotes March, June, September, December. The 1$^{st}$, 2$^{nd}$, 3$^{rd}$, 4$^{th}$ column denotes the two-way coupled run (OCNCPL), the one-way coupled run (OCNDYN), the standalone MITgcm run (OCNSTA) and the observations (OSISAF).

Reference:

Bromwich, D. H., K. M.Hines, and L.-S.Bai, 2009: Development of Polar Weather Research and Forecasting Model: 2. Arctic Ocean. J. Geophys. Res., 114, D08122, doi:10.1029/2008JD010300.

Chen, F., and J.Dudhia, 2001: Coupling an advanced land surface-hydrology model with

the Penn State–NCAR MM5 modeling system. Part I: Model implementation and sensitivity. Mon. Wea. Rev., 129, 569–585.

Hansen, J., G. Russell, D. Rind, P. Stone, A. Lacis, S. Lebedeff, R. Ruedy and L.Travis, 1983: Efficient Three-Dimensional Global Models for Climate Studies: Models I and II. Monthly Weather Review, 111, 609 – 662.

Liang, X. and M. Losch, On the Effects of Increased Vertical Mixing on the Arctic Ocean and Sea Ice. Journal of Geophysical Research: Oceans, 2018. 123(12): p. 9266-9282.

Liang, X., et al., Using Sea Surface Temperature Observations to Constrain Upper Ocean Properties in an Arctic Sea Ice-Ocean Data Assimilation System. Journal of Geophysical Research: Oceans, 2019. 124(7): p. 4727-4743.

Liang, X., et al., Evaluation of ArcIOPS sea ice forecasting products during the ninth CHINARE-Arctic in summer 2018. Adv. Polar. Sci., 2020, 31(1), 14-25.

Sun, R., Subramanian, A. C., Miller, A. J., Mazloff, M. R., Hoteit, I., and Cornuelle, B. D.: SKRIPS v1.0: a regional coupled ocean–atmosphere modeling framework (MITgcm–WRF) using SMF/NUOPC, description and preliminary results for the Red Sea, Geosci. Model Dev., 12, 4221-4244, 10.5194/gmd-12-4221-2019, 2019

Mu, L., et al., Arctic Ice Ocean Prediction System: evaluating sea-ice forecasts during Xuelong's first trans-Arctic Passage in summer 2017. Journal of Glaciology, 2019. 65(253): p. 813-821.

Winton, M, 2000: A reformulated Three-layer Sea Ice Model. Journal of Atmospheric and Ocean Technology, 17, 525 – 531.

Van Pham, T., et al., New coupled atmosphere-ocean-ice system COSMO-CLM/NEMO:

assessing air temperature sensitivity over the North and Baltic Seas. Oceanologia, 2014. 56(2): p. 167-189.

---

## Author Response (AR1)

Reply to Reviewer Comment 1 (RC1)

*Comment:*
*This manuscript describes a coupled via C-Coupler2 Arctic ocean sea ice configuration of MITgcm and Polar WRF atmosphere. The model is intended for high quality Arctic sea ice seasonal predictions. There is large demand for high quality regional climate models of the Arctic basin and such activity must be strongly appreciated. In the manuscript the setup has been validated for year 2012 because of a strong storm formed off the coast of Alaska on 5 August 2012. The role of sea ice-ocean-atmosphere interaction has been addressed. Although the authors demonstrate good modelling skills, good knowledge of the Arctic Ocean system and impressive level of model validation the paper in it's resent form failed to convince me that the new climate model is ready to use and that is better than any existing global climate model. My main criticism is that (1) the performance of the model on different HPC systems, regarding scalability and the costs of all individual components is not addressed, (2) the presented configuration is not properly tuned. I encourage this paper for resubmission after these weaknesses have been fixed.*

Reply:

The authors thank the reviewer for the insightful comments, and we completely agree with the questions and comments raised by the reviewer, which have helped us to improve the quality of the manuscript. We have carefully considered the reviewer's comments. Some paragraphs are rewritten and figures are redrawn. Regarding to the two main criticisms, we have added a paragraph of model scalability and the costs of all individual components in the revised manuscript. We also tried different sea ice albedo parameters in the MITgcm to get a better sea ice simulation and re-written Section 4 with new results. To further adequately address the two main criticisms, our replies are as follows:

1. Following the comments on the model scalability and the costs, we run several experiments using different CPU numbers, each experiment runs for 7 model days, and we obtain the following results:

Table 1: Comparison of CPU time spent on coupled and stand-alone runs. The CPU time spent on two stand-alone simulations are presented to show the difference between coupled and stand-alone simulations. 'total_cpu_number' denotes the requested CPUs, 'total_run_time' denotes the total CPU elapsed time. 'wrf_interface', 'wrf_integration', 'mitgcm_interface' and 'mitgcm_integration' denote the CPU elapsed time used for coupling interface by the WRF, numerical integration by the WRF, coupling interface by the MITgcm, and numerical integration by the MITgcm, respectively. 'wrf_time_alone' denotes the CPU elapsed time of the stand-alone WRF runs. 'mitgcm_time_alone' denotes the CPU elapsed time of the stand-alone MITgcm runs. Each run is integrated for 7 model days.

| total_cpu_number | cpu_number on_each_component_model | total_run_time (unit: s) | wrf_interface (unit: s) | mitgcm_interface (unit: s) | wrf_integration (unit: s) | wrf_time_alone (unit: s) | mitgcm_integration (unit: s) | mitgcm_time_alone (unit: s) |
|---|---|---|---|---|---|---|---|---|
| 28 | 14 | 12840 | 4.8 | 12131 | 12835.2 | / | 709 | / |
| 56 | 28 | 12000 | 4.74 | 11196 | 11995.26 | 7140 | 804 | 317 |
| 112 | 56 | 10440 | 5.16 | 6477 | 10434.84 | 3960 | 3963 | 154 |
| 224 | 112 | 3780 | 5.26 | 3550 | 3774.74 | 2160 | 230 | 96 |
| 448 | 224 | 2460 | 5.21 | 2116 | 2454.79 | 1560 | 344 | 68 |
| 896 | 448 | 1380 | 358 | 48 | 1022 | 1320 | 1332 | 84 |

In our model configuration, the requested total CPUs are assigned equally to the component models, that is, if we request 448 CPUs, then 224 CPUs are assigned to the WRF and 224 CPUs are assigned to the MITgcm. The total_run_time of the coupled model decreases from 12840 s to 1380 s when total_cpu_number increases from 28 to 896. Limited by computational resource of our center, we can not perform experiment which uses more than 1000 CPUs.

In the above table, the "wrf_interface" expresses time of coupling process implemented by the WRF, then the time of integration process implemented by the WRF, i. e. "wrf_integration" can be calculated as : "total_run_time" minus "wrf_interface". The "mitgcm_interface" expresses time of coupling process implemented by the MITgcm, then the time of integration process implemented by the MITgcm, i. e. "mitgcm_integration" can be calculated as : "total_run_time" minus "mitgcm_interface". The "wrf_time_alone" expresses runtime of the standalone WRF which implements 7 model days integration. The "mitgcm_time_alone" expresses runtime of the standalone MITgcm which implements 7 model days integration.

We find that, when total_cpu_number is not larger than 448, the "mitgcm_integration" is substantially smaller than the "wrf_integration", meaning that the efficiency of the coupled model depends on the WRF component model. When total_cpu_number is larger than 448, the efficiency of the coupled model depends on the MITgcm component model. Additionally, both the integration efficiency of the component models in the coupled model are lower than those of the standalone model runs. The decrease in parallel efficiency results from the increase of communication time, load imbalance, and I/O (read and write) operation per CPU core (Christidis, 2015). By comparing the time cost of stand-alone WRF and MITgcm integration the parallel efficiency of the coupled model is higher than both ocean-alone or atmosphere-alone models with same numbers of grid points per CPU core.

[Figure]

Fig.R1 The parallel efficiency (left) and speed-up (right) test of the coupled model, employing up to 896 CPU cores. The simulation using 28 CPU cores is regarded as the baseline case when computing the speed-up.

Table R1 added to Table 1 in the revised manuscript
Fig.R1 added to Figure 4 in the revised manuscript
Line 111 :"In section 3, a scalability test of the coupled model is performed to investigate its parallel capability." added
Line 206-229 :"In this section, the parallel efficiency of the ArcIOAM is investigated. Different numbers of CPU cores are used to evaluate the parallel speed-up of the coupled model. The CPU elapsed time spent on coupling interface of each component model in the coupled runs are detailed. Additionally, the parallel efficiency of each component model in the stand-alone runs are calculated for references. The parallel efficiency tests are performed on the High performance computing cluster at NMEFC. The High performance computing cluster is a Lenovo Blade Server system composed of 240 dual-socket compute nodes based on 14-core Intel Haswell processors running at 2.4 GHz. Each node has 128GB DDR4 memory running at 2133 MHz. Overall the system has a total of 6270 CPU cores (240 nodes x 2 x 14 CPU cores) and has a theoretical peak speed of 258 tetaflops. The parallel efficiency of the scalability test is $N_{p0} t_{p0}/ N_{pn} t_{pn}$, where $N_{p0}$ and $N_{pn}$ are the number of CPUs employed in the base case and the test case, respectively; $t_{p0}$ and $t_{pn}$ represent the CPU elapsed time in the base case and the test case. The speed-up is defined as $t_{p0}/t_{pn}$, which is the relative improvement of the CPU time. The scalability tests are performed by integrating 7 model days for the stand-alone Polar WRF, the stand-alone MITgcm and the coupled runs.

In the ArcIOAM runs, the requested CPUs are assigned equally to the component models. The minimum CPUs we use is 28, i. e. $N_{p0}$ = 28. Limited by computational resource, the maximum CPUs we can use is 896. The total CPU elapsed time in the coupled runs decreases from 12840 s to 1380 s when the requested CPUs increases from 28 to 896 (Table 1). When the requested CPUs are not larger than 448, the CPU elapsed time used for numerical integration by the MITgcm is substantially smaller than that for numerical integration by the WRF, meaning that the efficiency of the coupled model depends on the WRF component model. When the requested CPUs are larger than 448, the efficiency of the coupled model depends on the MITgcm

component model.

The parallel efficiency of the coupled model remains more than 90% when employing less than 112 cores and is still as high as 80% when using 224 cores (Figure 4). The parallel efficiency of the stand-alone MITgcm is near to that of the stand-alone Polar WRF when the requested CPUs are less than 448, while both of them are substantially lower than the coupled model. The parallel speed-up of the coupled model is higher than the stand-alone component model. The decrease in parallel efficiency results from the increase of communication time, load imbalance, and I/O (read and write) operation per CPU core (Christidis, 2015). " added

2. In the MITgcm, sea ice albedo is a function of four kinds of ice/snow surface type: dry ice, wet ice, dry snow, wet snow. According the reference (Nguyen et al., 2011, Arctic ice ocean simulation with optimized model parameters: Approach and assessment, JGR-oceans), typical range of sea ice albedo in the AOMIP (Arctic Ocean Models Intercomparison Project) is: 0.6-0.75 for dry ice, 0.5-0.68 for wet ice, 0.8-0.84 for dry snow, 0.6-0.77 for wet snow. Typical sea ice albedo in the MITgcm under the JRA25 forcing is: (dry ice: 0.7, wet ice: 0.71, dry snow:0.87, wet snow: 0.81). They also found the optimized sea ice albedo parameters depend on the selected atmospheric forcing. In our previous studies (Liang et al., JGR-oceans, 2018, 2019), we use the sea ice albedo of (dry ice: 0.75, wet ice: 0.7, dry snow:0.86, wet snow: 0.8) when the JRA55 forcing is used. For this study using CFSR forcing, before manuscript submission, we tested the above albedo parameters for the coupled model and found that the model produced more sea ice than the observation. So we reduced the albedo parameters and tested the group of (dry ice: 0.65, wet ice: 0.55, dry snow:0.8, wet snow: 0.7) for the coupled model, and we found this group of albedo parameters is appropriate for the coupled model when the CFSR forcing is used.

Besides, we add the standalone MITgcm simulation in 2012 for the comparison. To keep consistency between the coupled model and standalone MITgcm model, the standalone MITgcm simulation is forced by surface variables derived from the CFSR data and uses the same albedo group of (dry ice: 0.65, wet ice: 0.55, dry snow:0.8, wet snow: 0.7). Results of the modeled sea ice show that the two-way coupled model generates more rational sea ice distribution than the standalone MITgcm run. The modeled and observed monthly sea ice concentration are shown as follows:

[Figure]

Fig.R2 Monthly mean sea ice concentration in 2012. The 1$^{st}$, 2$^{nd}$, 3$^{rd}$, 4$^{th}$ row denotes March, June, September, December. The 1$^{st}$, 2$^{nd}$, 3$^{rd}$, 4$^{th}$ column denotes the two-way coupled run (OCNCPL), the one-way coupled run (OCNDYN), the standalone MITgcm run (OCNSTA) and the observations (OSISAF).

We have re-drawn Figure4-10 by adding results of stand-alone experiment. And we also added three figures showing the deviation of three experiments minus observations.

Detailed replies to specific comments by the reviewer are presented below:

*Comment :*
*In the model description, there need to be discussion on why fields and not fluxes are coupled. Do the authors guarantee the same bulk formulas are employed on the atmospheric and ocean sides? Which difference is expected if the fluxes are coupled? I guess that COSMO-CLM/NEMO group has some experience with it although not with the Arctic region. I find this aspect is more important than describing the computation of the corner geographic information for MITgcm. The latter piece I*

*would even omit due to its simplicity. A following chapter after the model description, which gives more information about the model scalability and cost is required.*

Reply:

In the original manuscript, we have cited the paper from COSMO-CLM/NEMO group (Van Pham et al. 2014). For COSMO-CLM/NEMO model, the exchanged fields from COSMO-CLM to NEMO are the flux densities of water, momentum, solar radiation, non-solar energy and sea level pressure; and from NEMO to COSMO-CLM are SST and the fraction of sea ice. Regarding to the MITgcm model configuration we used, instead of forcing the model with heat flux data, the model calculates these fluxes using the changing sea surface temperature and ice surface temperature. We need to read in some atmospheric data: 2 m air temperature, 2 m air humidity, downward shortwave radiation, downward longwave radiation, precipitation, 10 m wind speed. This combination of setups have been used in our ocean-seaice model for several years and showed reasonable results in aspects of sea ice forecasts (Liang et al., JGR-oceans, 2018, 2019). Therefore when we build the coupled model, the same setups from the standalone ocean-seaice model are kept. We have checked the manual and source code from both the WRF and the MITgcm.

(1) The heat fluxes calculated in the MITgcm are shown below (https://mitgcm.readthedocs.io/en/latest/phys_pkgs/bulk_force.html):

Sensible heat flux (Qs):

$$Qs = \rho_{air} cp_{air} u^* T^*$$

Latent heat flux (Ql):

$$Q_l = \rho_{air} u^* q^*$$

Where

$$u^* = c_u u_s$$

$$T^* = c_T \Delta T$$

$$q^* = c_q \Delta q$$

$$c_u = c_T = c_q = \frac{\kappa}{\ln(z_{ref} / z_{rou})}$$

$\rho_{air}$ : air density, $cp_{air}$ : specific heat at constant pressure, $u_s$ : wind speed, $\kappa$ : Von Karman constant, $z_{ref}$ : reference height and $z_{rou}$ : roughness length scale which could be a function of type of surface.

(2) The heat fluxes calculated in the WRF are shown below (http://www2.mmm.ucar.edu/wrf/users/docs/user_guide_V3.8/contents.html) :

Sensible heat flux (H):

$$H = \rho \, c_p u_* \theta_*$$

Latent heat flux (E):

$$E = \rho \, u_* q_*$$

$$u_* = \frac{kV_r}{\ln(\frac{z_r}{z_0}) - \psi_m}$$

$$\theta_* = \frac{k\Delta\theta}{\ln(\frac{z_r}{z_{0h}}) - \psi_h}$$

$$q_* = \frac{k\Delta q}{\ln(\frac{z_r}{z_{0q}}) - \psi_h}$$

Where subscript $r$ is reference level (the lowest model level, or 2 m or 10 m), $\Delta$ refers to difference between surface and reference level value, $z_0$ are the roughness lengths, $k$ is the von Karman constant.

(3) The above calculations of bulk formula for sensible heat flux and latent heat flux are almost same. To further prove this, we also compare the sensible and latent flux from the WRF output and the MITgcm output within the coupled model based on the result on March 1, 2012. The results show very little discrepency, which mainly because that parameter setup is slightly different from atmosphere and ocean sides. In future work, we will use fluxes as exchange variables instead of fields, ensuring the energy balance.

Line 195-201 :"Normally in coupled models the coupler controls the exchange of heat and momentum fluxes among component models. In our model configuration, instead of coupling fluxes directly, we use the C-Coupler2 to control the exchange of fields between the Polar WRF and the MITgcm. Heat and momentum fluxes are calculated separately in each component model. Both the Polar WRF and the MITgcm use the same Bulk Formula and almost same parameters in calculating fluxes, which guarantees the quasi-conservation of heat and momentum transmission between the component models. The bilinear interpolation algorithm is involved in the transmission of model variables between the horizontal grid of the Polar WRF and that of the MITgcm." added

We agree with the reviewer that the computation of the corner geographic information for the MITgcm is more like a technical issue and with simplicity. So the part of the computation of corner information and Figure 3 are removed from the revised manuscript. We have already added a new chapter of model scalability and

cost analysis in the revised manuscript.

Line 229-243: removed
Figure 3 removed

[Figure]

Fig.R3 Sensible and latent heat fluxes derived from WRF and MITgcm output on March 1, 2012. Differences (WRF - MITgcm) of heat fluxes are calculate by interpolation to the same grid.

*Comment:*
*I believe that C-Coupler is a good tool to use but the statement that a model produces bitwise identical results with a different coupler means that the coupler just works. Is it better in terms of performance? Which interpolation option do you use (question is more relevant for the wind stress)? As an illustration it would be good to see the curl of the wind stress on the atmospheric and oceanic meshes (instead of Fig.3).*

Reply:
The innovations of C-Coupler are flexible and automatic coupling configuration and 3-D coupling capability, which is easier for users to build coupled models. For the interpolation (remapping) from a source horizontal grid to a target horizontal grid, users can use the remapping weights that are either automatically generated by C-Coupler2 in parallel, or read from an existing remapping weight file produced by an external software tool such as SCRIP, ESMF, YAC, CoR, etc. Remapping configuration files enable to flexibly and conveniently specify how to remap coupling fields between grids. For our model, we used the default remapping configuration, the bilinear remapping algorithm is used for remapping the "state" fields between the horizontal grids.

The momentum variable transferred in our model is 10m-wind instead of wind stress. We have calculated the wind stress in WRF and made a comparison with MITgcm

output on March 1, 2012. The results show very little discrepency, which mainly because of parameter setup is slightly different from atmosphere and ocean sides.

[Figure]

Fig.R4 Wind stress curl (unit: Nm$^{-2}$) derived from (a) MITgcm and (b) WRF output on March 1, 2012. Difference (WRF - MITgcm) of wind stress curl is calculate by interpolation to the same grid.

Line 200-203 :"Figure 3 shows wind stress curl derived from the Polar WRF output and the MITgcm output, as well as their difference on March 1, 2012. It can be seen that the Polar WRF and MITgcm model generate similar wind stress curl pattern, and the difference due to interpolation algorithm and momentum calculation accounts for less than 5% of the wind stress curl (Figure 3c)." added
Figure R4 added to Figure 3 in the revised manuscript

*Comment:*
*In the OCNDYN run is it just setting alpha to 1.0? I believe that one year of coupled simulation is too short to validate the model. Either ensemble of runs or a longer simulation is desired here. I don't want to force the authors to do much of additional work but the cheapest way would be to add the stand-alone MITgcm run for the comparison here. The spin up with JRA55 has been already computed.*

Reply:
For the OCNDYN run , we switch off the export interface of coupling in the code to close the variable transfer from ocean to atmosphere. The alpha in section of coupling strategy is a relax coefficient weight to combine the boundary variable, to diminish the abrupt value changes from two sources. We agree that one year is short to validate the model. So we add the standalone MITgcm runcase for the comparison. The results are shown in the revised manuscript.

*Comment:*
*Section 4.1, which introduces Fig. 5 illustrates that the model has not been tuned properly yet. Although the authors (line 254) claim that OCNCPL is closer to observations, which might be true, but I see that both runs failed. Here it is important to give (at least visually) the measure of the error. A stand-alone MITgcm run, hence, would be a good choice. Provided the high skill of validation made in the following chapters I assume that the model has to be better tuned first.*

Reply:

We have already tried different seaice parameters and tuned the model. The appropriate albedo of (dry ice: 0.65, wet ice: 0.55, dry snow: 0.8, wet snow: 0.7) in the coupled model for the CFSR forcing is used. Compared with the one-way coupled and standalone run, the OCNCPL case shows better results. We re-write the section 4 of validation with new results. The measure of the sea ice concentration error between the model and observation are shown as follows:

[Figure]

Fig.R5 Sea ice concentration bias between the model and the OSISAF observation in 2012. The 1st, 2nd, 3rd, 4th row denotes March, June, September, December. The 1st, 2nd, 3rd column denotes the two-way coupled model, the one-way coupled model, the standalone MITgcm run.

Figure R5 added to Figure 7 in the revised manuscript

*Comment:*
*Minor things: line 181: "...without any data assimilation. . .". There was nothing said about new model is within an assimilation framework before or after. Why to mention this?*

Reply:
In our institute, we have established an Arctic Ice Ocean Prediction System (ArcIOPS) based on the MITgcm and ensemble Kalman Filter data assimilation algorithm to carry out operational Arctic synoptic-scale sea ice forecast. Our future plan is to implement the coupled model with data assimilation algorithm to carry out Arctic seasonal sea ice prediction.

Line 246-247: "As a starting point, we need to evaluate the Arctic coupled model performance on seasonal timescale without any data assimilation." Revised to "As a starting point, we need to evaluate its performance on seasonal timescale."

*Comment:*
*line 182: "...the coupled model free simulations..." what do you mean?*

Reply:

Free simulation is aiming at the experiment with data assimilation in planning.

Line 248: "In this work, we perform the coupled model free simulations in the year of 2012 with special focuses on the summertime." Revised to "In this work, we perform the coupled model simulations in the year of 2012"

*Comment:*
*line 188: indeed, nothing about atmosphere. Patterns of SLP although would be of interest.*

Reply:
We have compared the patterns of atmospheric variable (2m temperature, SLP and wind fields) in OCNCPL run and OCNDYN run.

[Figure]

Fig.R6 Bias of 2m temperature, SLP and 10m wind speed from OCNCPL and OCNDYN on September 8, 2018.

*Comment:*
*line 230: too speculative. did you do another run with different albedo?*

Reply:
We have tried different sea ice albedo combinations and then chose a best albedo combination for the CFSR forcing. In our previous studies (Liang et al., JGR-oceans, 2018, 2019), we use the albedo combination of (dry ice: 0.75, wet ice: 0.7, dry snow:0.86, wet snow: 0.8) when JRA55 forcing is used. For this study using CFSR forcing, we tested the above albedo combination for the coupled model and found that the model produces more sea ice than the observation. So we reduced the albedo parameters and tested the combination of (dry ice: 0.65, wet ice: 0.55, dry snow:0.8, wet snow: 0.7) for the coupled model, and we found this group of albedo parameters is appropriate for the CFSR forcing.

Line 306-307: "A series of sensitivitiy experiments are performed to get an optimal combination of sea ice parameters (figures not shown)." Added

*Comment:*
*line 290: I would elaborate more on this if possible.*

Reply:
Line 385-390: "Day et al. (2014) pointed out that sea ice behaves long-term memory of melting-freezing processes. Notz and Bitz (2017) indicated that summertime sea ice thickness has an important influence on sea ice state in the following spring through the ice thickness-ice growth feedback. A negative anomaly of sea-ice area in late summer induces larger heat losses in autumn and winter from the ocean to the atmosphere due to enhanced outgoing long-wave radiation and turbulent heat fluxes, this causes thinner snow and ice due to later freeze-up and hence larger heatconduction fluxes through sea ice, eventually leading to larger ice-growth rates."
Added

*Comment:*
*line 366: not really or show that it is better than in other (global) models*

Reply:
Indeed, we did not compare our results to other models. The motivation of our work is to enhance the operational sea ice seasonal prediction by coupling atmosphere, ocean and sea ice. Sea ice model intercomparison is a good index to weigh the performance of coupled models. In future, we hope that we can join such international project to evaluate and further improve our models.

*Comment:*
*line 372: is it due to the albedo feedback? In OCECPL it is computed on the MITgcm side. What happens in OCEDYN? Again, figures 6c, 7 would have more value if the model is tuned. In the present form the model is not ready for this validation.*

Reply:
We believe that two-way coupling between the WRF and the MITgcm provides a more rational representation of real air-ice-ocean physical processes, which includes the important ice-albedo feedback in early summer. In the MITgcm, sea ice albedo is calculated based on several variables, such as snow depth on ice, ice surface temperature. In the OCNCPL run, albedo is a coupling variable which affects both the WRF and the MITgcm. In the OCNDYN run, albedo used in the WRF are directly read from the CFSR forcing data.

In our original manuscript, we make a mistake when calculate sea ice extent in Figure 4a and 4b, which leads to the negative sea ice extent bias in Figure 4a. We confuse sea ice area with sea ice extent. Actually the blue and red curves in Figure 4a and 4b in the original manuscript are sea ice area. We redraw the modeled sea ice extent and add the standalone MITgcm run for the comparison (see the following figure). Compared with the standalone MITgcm run, the modeled sea ice extent in the coupled runs are more closer to the observation. With respect to the one-way coupled run, the spatial distribution of summertime sea ice concentration in the two-way coupled run is more closer to the OSISAF observation.

Line 499-503: "The two-way coupling between the Polar WRF and the MITgcm provides a more rational representation of real air-ice-ocean physical processes, which includes the important ice-albedo feedback in early summer. In the MITgcm, sea ice albedo is calculated based on several variables, such as snow depth on ice, ice surface temperature. In the OCNCPL run, albedo is a coupling variable which affects both the Polar WRF and the MITgcm. In the OCNDYN run, albedo used in the Polar WRF is directly read from the CFSR forcing data." Added

[Figure]

Fig.R7 Time series of (a) sea ice extent, (b) sea ice extent anomaly, and (c) root mean square error (RMSE) of modeled sea ice concentration with respect to the OSISAF observation in 2012. The black, red, blue and green lines in (a) denote sea ice extent of the MASIE observation, the OCNCPL run, the OCNDYN run, and the OCNSTA run respectively. The black, red, blue and green lines in (b) denote sea ice extent anomaly of the MASIE observation, the OCNCPL run, the OCNDYN run, and the OCNSTA run respectively. The red, blue and green lines in (c) denote the sea ice concentration RMSE of the OCNCPL run, the OCNDYN run, and the OCNSTA run, respectively.

Figure R7 added to Figure 7 in the revised manuscript

**Reply to Reviewer Comment 2 (RC2)**

*Comment:*
*This manuscript describes a "new" regional coupled model for the Arctic intended for short-term forecasting and shows some preliminary simulation results. My main issue with this manuscript is that I do not see the novelty here. Similar regional models such as ARCSyM (Lynch et al. 1995), RASM (Cassano, Maslowski, others), Rinke et al. 2000 have not been mentioned here. The "novel" aspect seems only to be the coupling of the MITgcm to the Polar WRF model and perhaps the new C-Coupler. I nearly recommended reject for this manuscript. It seems more like a technical report. Also, the English usage here is confusing and there are a number of grammatical errors that I do not have time to go into here. Here are some specific suggestions that might make this worthy of publication in GMD.*

Reply:
The authors thank the reviewer for the insightful comments. Thanks for pointing out that we omit similar works about regional coupled models. We have added these work as reference and re-written the introduction. The motivation of our work is to couple the Arctic ocean-seaice configuration of the MITgcm which is operationally running in our institute (ArcIOPS; Liang et al., 2020), to the Arctic atmospheric model (Polar WRF) in order to get a reasonable seasonal sea ice simulation. Beyond the scope of this work, our destination is operational sea ice seasonal prediction based on the coupled model and data assimilation algorithm. In my opinion, the novelty of our study is that we couple the Polar WRF and MITgcm which both featured with good performances in polar regions for the first time in the Arctic region and that we have solved some technical issues during the coupling process with a new coupler.

We have revised our manuscript by adding discussion on sea ice dynamic during coupling process. We have better motivated the manuscript in the introduction. We have proof-read the manuscript carefully to make it more readable.

Line 63-64: "The Arctic Region Climate System Model (ARCSyM) was developed to simulate coupled interactions among the atmosphere, sea ice, ocean, and land surface of the western Arctic (Lynch et al., 1995;Rinke et al., 2000)." Added
Line 68-73: "The Regional Arctic System Model (RASM) is a fully coupled, regional Earth system model covering the pan-Arctic domain (Maslowski et al., 2012;Cassano et al., 2017). The component models of RASM include the Weather Research and Forecasting (WRF) atmospheric model, the Variable Infiltration Capacity (VIC) land and hydrology model, and regionally configured versions of the ocean and sea ice models used in the Community Earth System Model (CESM): the CICE model and Parallel Ocean Program (POP)." Added
Line 81-84: "Since regional models can be run at higher resolution than global models, regional models can explicitly represent mesoscale features that may not be resolved in global models. Another potential advantage of the regional systems is that the

LBClateral boundary conditions can be controlled to get an optimal model input (Cassano et al., 2017)." Added

*Comment :*
*1. There needs to be more on the novel aspects here. What does this model provide specifically that previous models did not? What time scales is this intended for? Short-term forecasting of weeks? Seasons?*

Reply:
Sun et al. (2019) introduced a regional ocean-atmosphere coupled model covering the Red Sea based on the MITgcm and WRF. The novelty of our study is that we couple the Polar WRF and the MITgcm for the first time in the Arctic region and that we have solved some technical issues during the coupling process with a new coupler.
Both Polar WRF and MITgcm have specific features designed for polar regions, we speculate that coupling them will help us to improve seasonal sea ice prediction. We have used MITgcm model to make an operational Arctic synoptic-scale sea ice forecast for several years and showed reasonable results in aspects of sea-ice forecast in synoptic scale (ArcIOPS; Mu et al., 2019; Liang et al., 2020). This work is motivated by the need of a coupled Arctic sea ice-ocean-atmosphere model system for operational sea ice prediction of seasonal timescale (such as initialized at June and predict September sea ice).

Line 84-90: "This work is motivated by the need of a coupled Arctic sea ice-ocean-atmosphere model system for seasonal sea ice prediction in National Marine Environmental Forecasting Center of China." revised to "Aiming at operational seasonal sea ice prediction in the National Marine Environmental Forecasting Center (NMEFC) of China, the motivation of this work is to establish a fully coupled Arctic sea ice-ocean-atmosphere model with an capacity of reasonable sea ice simulation on seasonal timescale."

*Comment:*
*2. There is no mention of the land component. Is this just the imbedded Polar WRF component? Why is this not important to the Arctic simulations?*

Reply:
We apologize that we overlooked the description of land component in the section of Polar WRF. Yes, land component is embedded inside the Polar WRF. This Polar WRF incorporates many modifications to the standard version of the WRF. These adjustments are described by Bromwich et al. (2009), for example, adjustments to the surface parameterizations. The changes made in the Noah land surface model (LSM; Chen and Dudhia 2001) include using the latent heat of sublimation for calculating latent heat fluxes over ice surfaces, increasing the snow albedo and the emissivity value for snow, adjusting snow density, modifying thermal diffusivity and snow heat capacity for the subsurface layer, changing the calculation of skin temperature, and

assuming ice saturation in calculating the surface saturation mixing ratio over ice. Land component is absolutely important to the Arctic simulation, however at current stage, our coupled model has not capacity of coupling an individual land model, instead, we use the embedded land component in the Polar WRF for technical simplicity. We have added the introduction of land component in the section of Polar WRF description.

Line 138-143: "The Noah land surface model is embedded inside the Polar WRF. The changes made in the Noah land surface model (LSM;(Chen and Dudhia, 2001)) include using the latent heat of sublimation for calculating latent heat flux over ice surface, increasing the snow albedo and the emissivity value for snow, adjusting snow density, modifying thermal diffusivity and snow heat capacity for the subsurface layer, changing the calculation of skin temperature, and assuming ice saturation in calculating the surface saturation mixing ratio over ice." Added

Line 504-506: "Land component is absolutely important to the Arctic simulation, however at current stage, our coupled model has not the capacity of coupling an individual land model, instead, we use the embedded land component in the Polar WRF for technical simplicity." Added

*Comment:*
*3. The MITgcm sea ice component is quite old and simplistic. What is the albedo formulation on sea ice? What about on land? What about a sea ice thickness distribution?*

Reply:
There are two calculations of surface albedo provided in the MITgcm.
1) from LANL CICE model:

$$\alpha = f_s \alpha_s + (1 - f_s)(\alpha_{i_{min}} + (\alpha_{i_{max}} - \alpha_{i_{min}})(1 - e^{-h_i/h_\alpha}))$$

where $f_s$ is 1 if there is snow, 0 if not; the snow albedo, $\alpha_s$ has two values depending on whether $T_s$ < 0 or not; $\alpha_{i\min}$ and $\alpha_{i\max}$ are ice albedo for thin melting ice, and thick bare ice respectively, $h_i$ is snow depth, and $h_\alpha$ is a scale height.

2) From GISS model (Hansen et al 1983):

$$\alpha = \alpha_i e^{-h_s/h_a} + \alpha_s(1 - e^{-h_s/h_a})$$

where $\alpha_i$ is a constant albedo for bare ice, $h_s$ is snow depth, $h_a$ is a scale height and $\alpha_s$ is a variable snow albedo

$$\alpha_s = \alpha_1 + \alpha_2 e^{-\lambda_a a_s}$$

where $\alpha_1$ is a constant, $\alpha_2$ depends on $T_s$, $\alpha_s$ is the snow age, and $\lambda_a$ is a scale frequency.

In our coupled model, surface albedo from LANL CICE model is used. Land component is from atmospheric component of Polar WRF mentioned in last reply.

Line 129-131: "In order to parameterize a sub-grid scale distribution for sea ice thickness, the mean sea ice thickness in each grid can be split into as many as 7 thickness categories in the MITgcm sea ice model. In our coupled model for simplicity, we use 2 thickness categories: open water and sea ice." Added

*Comment:*
*4. The authors suggest that CMIP5 models represent sea ice in a simple way, but MITgcm is not any more complicated. A number of CMIP5 models used the Los Alamos Sea Ice Model (CICE) which is leading edge. What about CMIP6?*

Reply:
We apologize for the improper statement about sea ice representation in the CMIP5 models in the original manuscript. The sea ice physical mechanism is rather complicate in the CICE. We have replaced the improper words in the manuscript. We have also gone through the entire manuscript to revise other improper words.

Line 40-44: "Climate models, such as those involved in the Coupled Model Intercomparison Project Phase 5 (CMIP5), normally incorporate sea ice model in a relatively simple way, thus can be used to generate long-term sea ice outlook with low confidence on spatial distribution." revised to "For climate model, the current generation of global climate models (GCMs) comprising phase 5 of the Coupled Model Intercomparison Project (CMIP5) show some biases in sea ice extent and thickness (Stroeve et al., 2012;Shu et al., 2015)."

*Comment:*
*5. The grid staggering discussion and Figure 3 are really not necessary.*

Reply:
We agree with the reviewer that the grid staggering discussion and Figure 3 are not necessary. So the part of computation of corner information and Fig. 3 are removed from the revised manuscript.

Line 229-243 deleted
Figure 3 deleted

*Comment:*

*6. The case study they use is the year 2012. The boundary forcing for the atmosphere is from NCEP/NCAR re-analysis and for the ocean from the ECCO model. The sea ice extent is reasonable, but the sea ice volume is biased low compared to PIOMAS. I believe the problem here is the simplicity of the sea ice thermodynamics and lack of a subgridscale thickness distribution. I feel like the model might be tuned for 2012 and want to see how it performs for other years. Some comparison to IceSAT data would also be beneficial here.*

Reply:

We agree that one year is short to validate the model. So we add runcases of 2016 (for references, not shown in the manuscript), and add the standalone MITgcm run in 2012 for the comparison in the revised manuscript. To keep consistency between the coupled model and standalone MITgcm model, the standalone MITgcm simulation is forced by surface variables derived from the CFSR data and uses the same albedo parameters to the coupled model. As the IceSat data covers 2003-2008, we can not validate the sea ice thickness simulations of 2012 and 2016 with the IceSat data. Instead, we choose Cryosat2 and SMOS data to validate sea ice thickness.
In our original manuscript, we make a mistake when calculate sea ice extent in Figure 4a and 4b, which leads to the negative sea ice extent bias in Figure 4a. We confuse sea ice area with sea ice extent. Actually the blue and red curves in Figure 4a and 4b in the original manuscript are sea ice area. We redraw the modeled sea ice extent and add the standalone MITgcm run for the comparison (see the following figure). Compared with the standalone MITgcm run, the modeled sea ice extent in the coupled runs are more closer to the observation. With respect to the one-way coupled run, the spatial distribution of summertime sea ice concentration in the two-way coupled run is more closer to the OSISAF observation.

[Figure]

Fig.R8 Time series of (a) sea ice extent, (b) sea ice extent anomaly, and (c) root mean square error (RMSE) of modeled sea ice concentration with respect to the OSISAF observation in 2012. The black, red, blue and green lines in (a) denote sea ice extent of the MASIE observation, the OCNCPL run, the OCNDYN run, and the OCNSTA run respectively. The black, red, blue and green lines in (b) denote sea ice extent anomaly of the MASIE observation, the OCNCPL run, the OCNDYN run, and the OCNSTA run respectively. The red, blue and green lines in (c) denote the sea ice concentration RMSE of the OCNCPL run, the OCNDYN run, and the OCNSTA run, respectively.

Results of the modeled sea ice in 2016 are shown as follows:

The year of 2016 is selected because of the next anomalous sea ice extent minima event after 2012 appeared in 2016. We also conduct two-way coupled run, one-way coupled run, standalone MITgcm run for the year of 2016. The initial fields on 2016.1.1 of the MITgcm and the WRF model are derived from the standalone MITgcm simulation forced by JRA55 data and from the CFSR reanalysis data. Time evolution of modeled sea ice extent shows that the two-way coupled run produces more reasonable sea ice extent compared with the MASIE data, although the modeled September sea ice concentration of the two-way run in the Arctic Pacific section overmelts.

[Figure]

Fig.R9 Same to the above figure but for the year of 2016.

[Figure]

Fig.R10 Monthly mean sea ice concentration in 2016. The 1$^{st}$, 2$^{nd}$, 3$^{rd}$, 4$^{th}$ row denotes March, June, September, December. The 1$^{st}$, 2$^{nd}$, 3$^{rd}$, 4$^{th}$ column denotes the two-way coupled run (OCNCPL), the one-way coupled run (OCNDYN), the standalone MITgcm run (OCNSTA) and the observations (OSISAF).

**Additional modifications**

Line 4: "Bo Lin1" added

Line 13-14: "A newly developed coupler, C-Coupler2 (the Community Coupler 2),"revised to "In the ArcIOAM configuration, the Community Coupler 2 (C-Coupler2)"

Line 16-17 :"A scalability test is performed to investigate the parallelization of the coupled model." added

Line 17-21: "ArcIOAM is demonstrated with focus on seasonal simulation of the Arctic sea ice and ocean state in the year of 2012. The results obtained by ArcIOAM, along with the experiment of one-way coupling strategy, are compared with available observational data and reanalysis products." revised to "As the first step toward reliable Arctic seasonal sea ice prediction, the simulation results in the year of 2012 of the ArcIOAM implemented with two-way coupling strategy, along with one-way coupling strategy, are evaluated with respect to available observational data and reanalysis products."

Line 21 :"Besides, the standalone MITgcm run with prescribed atmospheric forcing is performed for references." added

Line 22-24: "From the comparison, results obtained from two experiments both realistically capture the sea ice and oceanic variables in the Arctic region over a 1-year simulation period." revised to "From the comparison, all the experiments simulate rational evolution of sea ice and ocean states in the Arctic region over a one-year simulation period."

Line 24: "The two-way coupled model" revised to "The two-way coupling strategy"

Line 25 : "result" added

Line 35: "can be" revised to "are"

Line 36 : "marginal seas" added

Line 47: "running operationally" revised to "operationally running"

Line 61: "process" revised to "processes"

Line 62: "study" revised to "studies"

Line 79-80: "the regional weather prediction system and wave prediction system" revised to "the regional weather and wave prediction system"

Line 101: "to this end," removed

Line 102: "specific areas" revised to "lateral boundaries"; "developed" revised to "designed"

Line 103-106: "After implementing ArcIOAM, we run seasonal simulation of Arctic sea ice and ocean state in 2012. The simulated variables of the Arctic ocean and sea ice are examined and validated against available observational data and reanalysis products." revised to "After implementing the ArcIOAM, we evaluate the model performance in the year of 2012 against available observational data. This year is selected because of the historical sea ice extent minimum record in the satellite era."

Line 109 : "Besides, a stand-alone MITgcm simulation with prescribed atmospheric forcing is performed for references." added

Line 111: "Section 3" revised to "Section 4"

Line 112: "Section 4" revised to "Section 5"

Line 133: "the" revised to "an"

Line 142-144: "Two key modifications for the Polar WRF are optimization of surface energy balance and heat transfer for the Noah land surface model over sea ice and permanent ice surfaces, and" revised to "Other modifications for the Polar WRF include"

Line 155: "The Coupler Component Model" revised to "The Coupler"

Line 156: "use" revised to "have implemented"

Line 181: "1)" removed; "there is" revised to "producing"

Line 245: "3 Numerical Experiments" revised to "4 Numerical Experiments"

Line 246: "Arctic coupled model" revised to "ArcIOAM"

Line 248-251: "The year of 2012 is chosen because an unusually strong storm formed off the coast of Alaska on 5 August 2012, and tracked into the center of the Arctic Basin where it lingered for several days and generated stronger sea ice-ocean-atmosphere interaction(Simmonds and Rudeva, 2012)." added

Line 254-256: "The year of 2012 is chosen because an unusually strong storm formed off the coast of Alaska on 5 August 2012, and tracked into the center of the Arctic Basin where it lingered for several days and generated stronger sea ice-ocean-atmosphere interaction(Simmonds and Rudeva, 2012)." removed

Line 257: "abilities" revised to "capabilities"

Line 258: "simulations" revised to "variables"

Line 259: "Two" revised to "Three"

Line 264-266: "$\alpha$ in Equ. 1 is set to 1 in the OCNDYN run. The third experiment of

OCNSTA represents the stand-alone MITgcm simulation with the same sea ice albedo parameters to the coupled model but prescribed atmospheric forcing to keep consistency with previous two coupling experiments." added

Line 266-267: "the two runs" revised to "these cases"; "on" revised to "in"; "and sea ice" removed

Line 269: "used in the OCNCPL and OCNDYN runs, and the prescribed atmospheric forcing used in the OCNSTA run" added

Line 275: "the ice is" revised to "the zones of sea ice are"

Line 275-276: "The ocean and sea ice initial condition" revised to "The initial condition of ocean and sea ice"

Line 276-277: ". The stand-alone MITgcm simulation was" removed

Line 279: "in our previous study" removed

Line 282: "coupled" removed; "and used in our analysis" removed

Line 283: "4" revised to "5"

Line 284: "4" revised to "5"

Line 285-286: "The lowest Arctic sea ice extent in the satellite-observed era occurred in the summer of 2012" revised to "The Arctic sea ice extent minimum value appeared in the summer of 2012 in the satellite era"

Line 287-288: ", obtained from http://nsidc.org/data/masie/, in the year of 2012" removed

Line 291-292: "Compared with the OCNSTA run, results from the experiments with

coupling (OCNCPL and OCNDYN) are closer to observations. It is noted that" added

Line 291-292: "Compared with the OCNSTA run, results from the experiments with coupling (OCNCPL and OCNDYN) are closer to observations. It is noted that" added

Line 293: "2" revised to "1.5";  "except" revised to "after"

Line 296: "Comparing" revised to "By comparing";

Line 297: "quite" removed

Line 300: "and phase" added; "of the two runs" revised to "in the OCNCPL and OCNDYN runs"

Line 301-303: "The sea ice extent bias between the model states and the observations likely arise from the sea ice model systematic bias which is induced by the choice of sea ice and snow albedo parameters in the two runs." revised to "While results of stand-alone run shows lag of sea ice melts and freezes in advance compared with the observations."

Line 305: "In the MITgcm sea ice model" revised to "In the sea ice model of MITgcm"

Line 308: "rationally amplifying albedo parameters or " removed ;  "a" removed

Line 310: "We compare the modeled sea ice concentration" revised to "The modeled sea ice concentration is compared"

Line 311-312: ".; obtained from http://osisaf.met.no/; product identifier: OSI-409." removed

Line 315-316: "After 1 month of model state adjustment, three experiments shows similar patterns that RMSE is lower in winter and spring than in summer and autumn" added

Line 317: "January" revised to "November"; "(Figure 6)" added; "coupling" added

Line 321-347: "To further clarify sea ice spatial distribution, we show the modeled and observed monthly mean sea ice concentration in July, August and September (Figure 5). In July, the modeled sea ice extent of the OCNCPL run is similar to that of the OCNDYN run, but the modeled sea ice concentration of the OCNCPL run is much lower than that of the OCNDYN run in thick multiyear ice zone near the Canadian Arctic Archipelago and in the southern Beaufort Sea (Figure 5a and Figure 5b). The satellite observations show that the OCNCPL run still overestimates sea ice concentration in the southern Beaufort Sea and the Laptev Sea (Figure 5c). In August, the modeled sea ice melts quickly in the Eurasian marginal seas in the two runs. Compared with the satellite observations (Figure 5f), the OCNDYN run overestimates sea ice concentration in the southern Beaufort Sea while underestimates sea ice concentration in the center Arctic basin (Figure 5e). The OCNCPL run simulates similar sea ice extent to the satellite observations but with lower concentration in the center Arctic basin (Figure 5d). In September, the modeled sea ice in the marginal sea ice zone melts out in the two runs. Although the two runs simulate almost same sea ice extent, due to rational representation of sea ice-ocean-atmosphere interaction in the OCNCPL run, the modeled sea ice distribution of the OCNCPL run is closer to the observations (Figure 5g and Figure 5i)." revised to "To further clarify sea ice concentration spatial distribution, we show the modeled and observed monthly mean sea ice concentration (Figure 6) and deviation of model results and observation (Figure 7) in March, June, September and December  In March when the Arctic Ocean is almost fully covered

by sea ice, the main source of discrepancy appears in sea ice edge zones in the Atlantic side (Figure 6a-c). In June, sea ice concentrations are overestimated in the Arctic marginal seas in the OCNCPL and OCNDYN runs (Figure 7d-e). The modeled sea ice concentration in the OCNSTA run is more closer to the observations (Figure 7f). In September, the modeled sea ice in the marginal sea ice zone melts out in all runs (Figure 6i-k). Compared with the satellite observations (Figure 6l), sea ice in the OCNSTA run overmelts in summertime which leads to an anomalous negative bias of sea ice concentration in the Arctic (Figure 7i), the two coupled runs overestimate sea ice concentration in the southern Beaufort Sea while underestimates sea ice concentration in the center Arctic basin (Figure 7g-h). Although the two coupled runs simulate similar sea ice extent patterns, due to rational representation of sea ice-ocean-atmosphere interaction in the OCNCPL run, the modeled sea ice distribution of the OCNCPL run is closer to the observations (Figure 6i and Figure 6l). In December, the situation is similar with that in March when sea ice dominates almost entire Arctic region."

Line 348: "4" revised to "5"

Line 353-355: "The OCNSTA run simulates more rational sea ice growth rate from January to May but systematic negative sea ice volume bias compared with the PIOMAS data. The sea ice volume in the OCNCPL and OCNDYN runs shows better results than that in the OCNSTA run from June to December." added

Line 356-358: "Both the two runs produce less sea ice volume than the PIOMAS data almost in a whole year of 2012, partly resulting from that our model underestimates sea ice extent (Figure 4a) without assimilating any observation." revised to "However, both the two coupled runs produce less sea ice volume than the PIOMAS data in most time of 2012, partly resulting from that our model underestimates sea ice extent (Figure 5a) without assimilating any observation."

Line 358: "However, it is" revised to "It"

Line 373: "mathematically" removed

Line 379-380: "Comparing with the CS2SMOS data, both runs produce rational sea ice thickness evolution" revised to "Comparing with the CS2SMOS data, both coupled runs produce more rational sea ice thickness evolution than stand-alone run"

Line 393: "two" revised to "three"; "the" revised to "mentioned";

Line 394-395: "Compared with the coupled runs, the sea ice thickness in the OCNSTA run shows larger bias in pack ice zone while smaller bias in marginal ice zone." added

Line 398-399: "OCNCPL run" revised to "coupled runs"; "model" revised to "the coupled runs";

Line 405: "(Figure 9)" added;

Line 406: "; obtained from http://www.whoi.edu/beaufortgyre/" removed;

Line 412-427: "Spatial distributions of monthly mean sea ice thickness in June, September, and December are shown in Figure 8. In June, almost the whole Arctic basin is still covered by thick ice, large sea ice thickness deviations between the two runs mainly appear around sea ice edge where sea ice-ocean-atmosphere interaction can impact significant influence on sea ice melting rate (Figure 8c). In September, accompanied by the change of sea ice concentration pattern when involving sea iceocean-atmosphere interaction, the modeled sea ice becomes thicker in the central Arctic while thinner in the area near the Greenland Island and in the southern Beaufort Sea (Figure 8f). As summertime sea ice thickness has strong effect on preconditioning the following wintertime sea ice thickness (Day et al., 2014), the modeled sea ice of the OCNCPL run is universally thinner than that of the OCNDYN run in December (Figure 8i)." revised to "Spatial distributions of monthly mean sea ice thickness and its bias with respect to available  CS2SMOS data in March, June, September, and December are shown in Figure 10 and Figure 11. In March and December, all three runs underestimate sea ice thickness in central Arctic, while overestimate sea ice thickness in marginal sea ice zone (Figure 11). In March, the OCNSTA run overestimates sea ice thickness in the Pacific sector of the Arctic Ocean and in the Baffin Bay (Figure 11e). The coupled runs overestimate sea ice thickness in the northern Barents Sea while underestimate sea ice thickness in the western Chukchi Sea (Figure 11a and Figure 11c). In December, compared with the OCNDYN run, the modeled sea ice thickness in marginal sea ice zone in the OCNCPL run is more closer to the CS2SMOS data (Figure 11b), partly due to the rational sea ice distribution at the beginning of freezing season, as summertime sea ice thickness has strong effect on preconditioning the following wintertime sea ice thickness (Day et al., 2014).";

Line 428: "4" revised to "5"

Line 438-451: "The SST RMSE of the two runs with respect to the GMPE data from July to September are shown in Figure 9. We do not show the time evolution of the SST RMSE in whole year because the two timeseries do not obviously diverge in the other months. In general, the SST RMSE of the OCNCPL run is smaller than that of the OCNDYN run in the summer of 2012, which means the SST simulation also improves when sea ice-ocean-atmosphere interaction is allowed in the model. Spatial patterns of modeled and observed SST in July, August and September are shown in Figure 10. The GMPE SST data is available in ice-free areas (Figure 10a, Figure 10e and Figure 10i). Comparing with the OCNDYN run, in July and August the modeled ocean surface of the OCNCPL run warms in Fram Strait, the Barents Sea, the Kara Sea and the Bering Strait while colds in the Baffin Bay, the Greenland Sea and the Laptev Sea (Figure 10d andFigure 10h). In September strong warming in the OCNCPL run appears in the southern Beaufort Sea (Figure 10l). These SST modifications induced by sea ice-ocean-atmosphere interaction not only lead to the reduction of the modeled ocean surface temperature bias, but also help to maintain a more rational sea ice spatial pattern." revised to "The SST RMSE of the three runs with respect to the GMPE data are shown in Figure 12. In general compared with the coupled runs, the SST RMSE in the OCNSTA run is smaller in the summertime but larger in the rest. Spatial patterns of the modeled and observed SST in March, June, September and December are shown in Figure 13. Deviation of the modeled SST and the GMPE SST observation is demonstrated in Figure 14. The GMPE SST data is available in ice-free areas (Figure 13d, Figure 13h, Figure 13l and Figure 13p). In March and June, the OCNSTA run produces a warmer sea surface in the Nordic Seas, which explains the positive SST bias from January to June in Figure 12 compared with the coupled runs. In September the SST RMSE in the OCNCPL run (Figure 12) arises from the strong negative bias in the southern Beaufort Sea and the

Baffin Bay (Figure 14g).";

Line 460: "Table 1" revised to "Table 2"

Line 467-469: "Figure 11" revised to "Figure 15"

Line 472: "5" revised to "6"

Line 475-478: "By coupling the Polar WRF and the MITgcm for the first time in Arctic region, a series of specific setup including data interpolation between different grids and relaxation algorithm in specific areas are designed." revised to "By coupling the Polar WRF and the MITgcm for the first time in Arctic region, a series of specific procedures including data interpolation between different grids and relaxation algorithm in lateral boundaries are designed. The parallel efficiency of the coupled model is also investigated."

Line 479: "new coupled model of" removed

Line 480: "coupled" revised to "coupling"; "and" removed

Line 481: "and stand-alone oceanic simulation (OCNSTA)" added

Line 482: "two" revised to "the two coupled";

Line 483-486: "From the comparison, results obtained from two experiments both realistically capture the sea ice and oceanic variables in the Arctic region over a 1-year simulation period. The two-way coupled experiment gives equal or better results compared with the one-way coupled experiment." revised to "From the comparison, results obtained from the two-way coupling experiment capture the sea ice and ocean evolution in the Arctic region over a 1-year simulation period. The two-way coupling experiment gives better results compared with the one-way coupling experiment and stand-alone oceanic simulation, especially in summertime."

Line 507-508: "seasonal scale simulation in" revised to "results for";

Line 521: "process understanding" revised to "better understanding of mechanism";

Line 533: "BL worked on the scalability test." added

Line 565-567: "Cassano, J. J., DuVivier, A., Roberts, A., Hughes, M., Seefeldt, M., Brunke, M., Craig, A., Fisel, B., Gutowski, W., Hamman, J., Higgins, M., Maslowski, W., Nijssen, B., Osinski, R., and Zeng, X.: Development of the Regional Arctic System Model (RASM): Near-Surface Atmospheric Climate Sensitivity, Journal of Climate, 30, 5729-5753, 10.1175/jcli-d-15-0775.1, 2017." Added

Line 573-574: "Christidis, Z.: Performance and Scaling of WRF on Three Different Parallel Supercomputers, High Performance Computing, Cham, 2015, 514-528," Added

Line 577-578: "Day, J. J., Hawkins, E., and Tietsche, S.: Will Arctic sea ice thickness initialization improve seasonal forecast skill?, Geophysical Research Letters, 41, 7566-7575, https://doi.org/10.1002/2014GL061694, 2014." Added

Line 582: "Dirk Notz, and Bitz, C. M.: Sea ice in Earth system models, in: Sea Ice, 304-325, 2017" Added

Line 626-627: "Liang, X., ZHAO, F., Li, C., ZHANG, L., and LI, B.: Evaluation of ArcIOPS sea ice forecasting products during the ninth CHINARE-Arctic in summer 2018, Advances in Polar ence, v.31;No.78, 19-30, 2020." Added

Line 636-637: "Lynch, A. H., Chapman, W. L., Walsh, J. E., and Weller, G.: Development of a Regional Climate Model of the Western Arctic, Journal of Climate, 8, 1555-1570, 10.1175/1520-0442(1995)008<1555:doarcm>2.0.co;2, 1995." Added

Line 663-664: "Rinke, A., Lynch, A. H., and Dethloff, K.: Intercomparison of Arctic regional climate simulations: Case studies of January and June 1990, Journal of Geophysical Research: Atmospheres, 105, 29669-29683, 10.1029/2000jd900325, 2000." Added

Line 675-676: "Shu, Q., Song, Z., and Qiao, F.: Assessment of sea ice simulations in the CMIP5 models, The Cryosphere, 9, 399-409, 10.5194/tc-9-399-2015, 2015." Added

Line 736-739: "Figure 3: Wind stress curl (unit: Nm-2) derived from (a) the MITgcm output, (b) the Polar WRF output, and (c) their difference on March 1, 2012. The difference of wind stress curl between the Polar WRF and MITgcm is calculated by interpolating the Polar WRF output onto the MITgcm grid." added

Line 743-746: "Figure 4: The parallel efficiency (left) and speed-up (right) test of the coupled model and the stand-alone component models, employing up to 896 CPU cores. The simulation using 28 CPU cores is regarded as the baseline case when computing the speed-up. The tests are performed on a Lenovo Blade Server system composed of 240 dual-socket compute nodes based on 14-core Intel Haswell processors." added

Line 750: "Figure 4" revised to "Figure 5"

Line 751: "green" added

Line 752: "the OCNSTA run" added

Line 753: "green" added

Line 754: "green" added; "the OCNSTA run" added

Line 755: "the OCNSTA run" added

Line 760-763: "Figure 5: Modeled and observed monthly mean sea ice concentration. The top, middle, and bottom panels show the July, August, and September sea ice concentration, respectively. The left, middle, and right panels show sea ice concentration of the OCNCPL run, the OCNDYN run, and the OSISAF observations. OSISAF = Ocean and Sea Ice Satellite Application Facility" revised to "Figure 6: Modeled and observed monthly mean sea ice concentration. From top to bottom panels show the March, June, September and December sea ice concentration, respectively. From left to right panels show sea ice concentration of the OCNCPL run, the OCNDYN run, the OCNSTA run and the OSISAF observations. OSISAF = Ocean and Sea Ice Satellite Application Facility."

Line 766-770: "Figure 7: Deviation between the modeled and observed monthly mean sea ice concentration. From top to bottom panels show the March, June, September and December sea ice concentration deviation respect to the OSISAF observations, respectively. The left, middle, and right panels show results of the OCNCPL run, the OCNDYN run, and the OCNSTA run. OSISAF = Ocean and Sea Ice Satellite Application Facility." added

Line 775-784: "Figure 6: Time series of (a) total sea ice volume, (b) spatial mean sea ice thickness, and (c) the RMSE of sea ice thickness with respect to the satellite-retrieved observations in 2012. The black, red, and blue lines in (a) denote total sea ice volume of the PIOMAS data, the OCNCPL run, and the OCNDYN run, respectively. The black, red, and blue dots in (b) denote sea ice thickness of the CS2SMOS observations, the OCNCPL run, and the OCNDYN run, respectively. The black bar in (b)

represents the observational uncertainties of the CS2SMOS data. The red and blue masks in (c) denote sea ice thickness RMSE of the OCNCPL run and the OCNDYN run with respect to the SMOS observations in thin ice (< 1 m) region (line), the Cryosat-2 observations (circle), the CS2SMOS observations (triangle), respectively. Model grid points without available observations are not taken into the sea ice thickness RMSE calculation. PIOMAS = Pan-Arctic Ice Ocean Modeling and Assimilation System; SMOS = Soil Moisture Ocean Salinity" revised to "Figure 8: Time series of (a) total sea ice volume, (b) spatial mean sea ice thickness, and (c) the RMSE of sea ice thickness with respect to the satellite-retrieved observations in 2012. The black, red, green and blue lines in (a) denote total sea ice volume of the PIOMAS data, the OCNCPL run, the OCNSTA run and the OCNDYN run, respectively. The black, red, green and blue dots in (b) denote sea ice thickness of the CS2SMOS observations, the OCNCPL run, the OCNSTA run and the OCNDYN run, respectively. The black bar in (b) represents the observational uncertainties of the CS2SMOS data. The red, green and blue masks in (c) denote sea ice thickness RMSE of the OCNCPL run, the OCNSTA run and the OCNDYN run with respect to the SMOS observations in thin ice (< 1 m) region (line), the Cryosat-2 observations (circle), the CS2SMOS observations (triangle), respectively. Model grid points without available observations are not taken into the sea ice thickness RMSE calculation. PIOMAS = Pan-Arctic Ice Ocean Modeling and Assimilation System; SMOS = Soil Moisture Ocean Salinity."

Line 788-790: "Figure 7: Time series of sea ice thickness at three positions: (a) (75 °N, 150 °W), (b) (78 °N, 150 °W), and (c) (74 °N, 140 °W). The red and blue lines denote sea ice thickness of the OCNCPL run and the OCNDYN run, respectively. The black solid and dashed lines denote sea ice thickness observations of the BGEP ULSs, which were deployed in the summers of 2011 and 2012. The black lines of the BGEP ULS observations have been smoothed with the gray bar representing the observational uncertainties. BGEP = Beaufort Gyre Exploration Project; ULS = upward-looking sonar."

revised to "Figure 9: Time series of sea ice thickness at three positions: (a) (75 °N, 150 °W), (b) (78 °N, 150 °W), and (c) (74 °N, 140 °W). The red, blue and green lines denote sea ice thickness of the OCNCPL run, the OCNDYN run and the OCNSTA run, respectively. The black solid and dashed lines denote sea ice thickness observations of the BGEP ULSs"

Line 798-800: "Figure 8: Monthly mean sea ice thickness. The top, middle, and bottom panels show the June, September, and December sea ice thickness, respectively. The left, middle, and right panels show sea ice thickness of the OCNCPL run, the OCNDYN run, and the deviation between them." revised to "Figure 10: Monthly mean sea ice thickness. From top to bottom panels show the March, June, September, and December sea ice thickness, respectively. From left to right panels show sea ice thickness of the OCNCPL run, the OCNDYN run, the OCNSTA run and the CS2SMOS data."

Line 803-806: "Figure 11: Deviation of the modeled monthly mean sea ice thickness and the CS2SMOS data. The top, middle, and bottom panels show sea ice thickness deviation of the OCNCPL run, the OCNDYN run and the OCNSTA run, respectively. The left and right panels show results in March and December." added

Line 810-812: "Figure 9: Time series of the RMSE of modeled SST with respect to the GMPE observations in summer of 2012. The red and blue lines denote the SST RMSE of the OCNCPL run and the OCNDYN run, respectively. GMPE = Group for High-Resolution Sea Surface Temperature Multi-Product Ensemble." revised to "Figure 12: Time series of the RMSE of modeled SST with respect to the GMPE observations in 2012. The red, blue and green lines denote the SST RMSE of the OCNCPL run, the OCNDYN run and the OCNSTA run, respectively. GMPE = Group for High-Resolution Sea Surface Temperature Multi-Product Ensemble."

Line 815-820: "Figure 10: Modeled and observed monthly mean SST. Rows 1 to 3 show the July, August, and September SST, respectively. Columns 1 to 4 show the SST of the GMPE observations, the OCNCPL run, the OCNDYN run, and the deviation between the OCNCPL and OCNDYN runs, respectively. GMPE = Group for High-Resolution Sea Surface Temperature Multi-Product Ensemble" revised to "Figure 13: Modeled and observed monthly mean SST. From top to bottom panels show the March, June, September and December SST, respectively. From left to right panels show the SST of the OCNCPL run, the OCNDYN run, the OCNSTA run and the GMPE observations, respectively. GMPE = Group for High-Resolution Sea Surface Temperature Multi-Product Ensemble."

Line 822-826: "Figure 14: Deviation of the modeled monthly mean SST and the GMPE SST data. From top to bottom panels show the March, June, September and December SST deviation, respectively. From left to right panels show the SST of the OCNCPL run, the OCNDYN run and the OCNSTA run, respectively. GMPE = Group for High-Resolution Sea Surface Temperature Multi-Product Ensemble." added

Line 829: "Figure 11" revised to "Figure 15"

Line 833-840: Table 1 added

Line 843: "Table 1" revised to "Table 2"

[revised manuscript text omitted]

---

## Author Response (AR2)

**Reply to the Comments from Topical Editor**

*Comments to the Author:*

*To the authors,*

*Your manuscript has improved greatly due to your substantial revisions. However, as the reviewer points out, the overall focus of the manuscript is perhaps skewed and could be improved by being turned closer to a strictly technical description of the model. In particular, point #4 of the reviewer should be addressed, either by explaining how this new model is better than CMIP models on metrics appropriate for CMIP models (i.e. long time scales), or do a more appropriate comparison to a high-resolution operational model.*

Reply:

The authors thank the editor for insightful comments and suggestions. We have replied the reviewer's comments point by point. We have clarified in the revised manuscript that focus of our paper is the technical description of ArcIOAM as a new tool to perform regional sea ice simulation and operational sea ice prediction on seasonal timescale. We have added more technical details of the coupled system, such as concurrent mode of model integration, physical parameterizations of each component and configurations of different experiments. Moreover, to avoid the confusion, we have gone through the entire manuscript and removed the language suggesting that it is a significant advance over the coupled model already out there.

**Reply to the Comments from Reviewer #2**

*Comment :*

*1. I appreciate the efforts the authors put in to address my earlier concerns. The albedo formulation though is not better than the delta-Eddington multiple scattering formulation used in CICE. So, again this is not a significant advance. (see my point 4 below)*

Reply:

Thank you very much for your comments. The albedo formulation algorithm in the MITgcm is not better than the delta-Eddington multiple scattering formulation used in CICE. We agree that this is not an advance. We added the statement "It is noticed that the albedo formulation in the MITgcm sea ice model is simple and straightforward. The CICE model provides a more sophisticated scheme for sea ice albedo calculation." into Line 479-480. We have also gone through the entire manuscript and removed the language suggesting that it is a significant advance over the coupled model already out there.

*2. The authors compare their model to CMIP5 models. CMIP6 model results have been available for sometime. I think it is important to highlight why they believe their sea ice simulation is better than CMIP6 models.*

Reply:

Models involved in CMIP6 represent state-of-the-art sea ice prediction and outlook on seasonal to longer timescales. Global coupled models involved in CMIP6 have innate advantage in predicting sea ice evolution on seasonal to longer timescales because that the interactions between polar and mid-latitudes are considered in the models. Our model is a regional model and its potential capacity of sea ice seasonal prediction severely depends on the lateral boundary conditions both on the ocean side and the atmosphere side. The motivation of this work is purely to introduce our regional coupled model as a new tool to potentially perform operational seasonal sea ice prediction in our institute (National Marine Environmental Forecasting Center of

China). We have not any intention to compare our model to the CMIP6 models, because regarding to the CMIP6 models, our model is a different tool for a different purpose.

We have clarified in the abstract, introduction and conclusion parts that the focus of our paper is to introduce the technical description of ArcIOAM as a new tool to perform regional sea ice simulation and operational sea ice prediction on seasonal timescale.

We added the statement "Climate models comprising phase 6 of the Coupled Model Intercomparison Project (CMIP6) represent state-of-the-art sea ice prediction and outlook on seasonal to longer timescales. " to Line 39-43.

We added the statement "The ArcIOAM is designed for seasonal sea ice prediction up to 6 months, while on longer timescale the regional model's capacity is expected to severely depend on the lateral boundary forcing data. Global coupled models, such as those involved in CMIP6, have innate advantages in sea ice prediction and outlook on seasonal to longer timescale because that the interactions between high- and mid-latitudes are considered. " to Line 464-467.

*3. The CMIP simulations were for the whole 20th century and the authors have only demonstrated performance of their model in two years (2012, 2016). Both anomalously low sea ice years. What about recent higher years? For example, 2013.*

Reply:

Our model is aiming at seasonal sea ice prediction with special focus on summertime. So we choose two anomalously low sea ice years to perform the model simulations. These two years are selected because the sea ice-air-ocean interaction is more active when more open water is exposed to atmosphere. The results show that our model has the capacity to reasonably simulate sea ice evolution in summertime. We have also performed the two-way coupling run for the year of 2017, which is the normal sea ice year. The result shows that the model can rationally simulate September sea ice distribution in 2017 (Figure S1 and S2).

[Figure]

Figure S1. Simulated sea ice concentration on 2017.09.01 by the ArcIOAM

[Figure]

Figure S2. Observed sea ice concentration on 2017.09.01 by the OSISAF

*4. My main concern is that this is just a description of a new tool that may be appropriate for short-term Arctic forecasting. This is barely demonstrated. I do not believe they can motivate this by saying this is "better" than CMIP models. It is a different tool for a different purpose where the high resolution regional formulation is more beneficial. The best comparison is the RASM model and they cannot say that this is even an advance over the RASM model. I was ready to suggest "reject", but I believe that if the authors can rework this manuscript as more of a technical description of their model and remove the language suggesting that it is a significant advance over what is already out there then it might be acceptable. I can understand the need to document their model system and perhaps GMD is the right place. However, I still believe this is not acceptable in it's current form.*

Reply:

Thanks for your comments and suggestions. We completely agree with the comments and suggestions raised by the reviewer.

We have clarified in the revised manuscript that the motivation of this work is the technical description of ArcIOAM as a new tool to perform regional sea ice simulation and operational sea ice prediction on seasonal timescale.

We have added more technical details of the coupled system, such as concurrent mode of model integration, physical parameterizations and details of each component and configurations of different experiments.

We have gone through the entire manuscript and removed the language suggesting that it is a significant advance over the coupled model already out there.

**Additional modifications**

Line 12-14: "Aiming at reliable Arctic sea ice prediction on seasonal timescale in the National Marine Environmental Forecasting Center of China, a new Arctic regional coupled sea ice-ocean-atmosphere model (ArcIOAM) has been established." added

Line 14 :"description and " added; "a new Arctic regional coupled sea ice ocean atmosphere model," removed

Line 34-35 :"(Bailey et al., 2020)" added

Line 39-43 :"For climate model, the current generation of global climate models (GCMs) comprising phase 5 of the Coupled Model Intercomparison Project (CMIP5) show some biases in sea ice extent and thickness (Stroeve et al., 2012;Shu et al., 2015)."

revised to "Climate models comprising phase 6 of the Coupled Model Intercomparison Project (CMIP6) represent state-of-the-art sea ice prediction and outlook on seasonal to longer timescales."

Line 79-83 :"Yang et al. (2020) has developed a coupled atmosphere-sea ice-ocean model configured for the pan-Arctic with the Coupled Ocean-Atmosphere-Wave-Sediment Transport modeling system (COAWST). A data assimilation system of ensemble Kalman filter is combined with this coupled model to assimilate satellite sea ice observations to improve initial sea ice conditions." added

Line 96-104 :"In coupled model systems, moisture, heat and momentum are often accomplished through the use of a separate coupling software like OASIS-MCT (Craig et al., 2017) or framework like the Earth System Model Framework (ESMF) (DeLuca et al., 2012) which links component models flexibly and controls the exchange and interpolation of coupling variables. The coupler, which can handle data interpolation and data transfer between different models and different grids, is the crucial part in the coupled systems. Using the ESMF and the National United Operational Prediction Capability (NUOPC), Sun et al. (2019) introduced a regional ocean-atmosphere coupled model covering the Red Sea based on the MITgcm (Marshall et al., 1997) and the WRF model (Skamarock et al., 2008)." moved to Line 85-92

Line 93 :"providing" added

Line 95-96 :"(ArcIOAM) as a new tool to perform regional sea ice simulation and operational sea ice prediction on seasonal timescale." added

Line 108 :"designed" revised to "implemented"

Line 109 :"and reanalysis" added

Line 115 :"and configurations" added

Line 120-122 :"The newly developed regional coupled modeling system of ArcIOAM is introduced in this section. The descriptions of individual model components and the coupling strategy with C-Coupler2 are presented below. Detailed options of physical parameterizations and model settings for the Polar WRF, MITgcm models and C-Coupler2 are summarized in Table 1." added

Line 177 :"In the ArcIOAM, the requested CPUs are assigned equally to the MITgcm and Polar WRF model." added

Line 179-181 :"The component models are running in concurrent mode (Figure 2), that is, the component models run on mutually exclusive sets of cores, if one component model finishes earlier than the other, its resources are idle and wait for the other component model." added

Line 181-182 :"At each coupling time step, data transfer from the MITgcm to the Polar WRF is executed when data transfer from the Polar WRF to the MITgcm is completed, and vice versa for choice." added

Line 189, 197 and 723: "Figure 2" revised to "Figure 3"

Line 211, 214 and 728: "Figure 3" revised to "Figure 4"

Line 229 and 820: "Table 1" revised to "Table 2"

Line 235 and 733: "Figure 4" revised to "Figure 5"

Line 251: "Table 3" added

Line 269: "specific" removed

Line 278, 288, 300, 331 and 741: "Figure 5" revised to "Figure 6"

Line 303, 309, 311, 314, 319, 320 and 751: "Figure 6" revised to "Figure 7"

Line 309, 312, 316, 317 and 756: "Figure 7" revised to "Figure 8"

Line 324, 335, 352, 367, 476 and 761: "Figure 8" revised to "Figure 9"

Line 377 and 773: "Figure 9" revised to "Figure 10"

Line 385 and 783: "Figure 10" revised to "Figure 11"

Line 385, 387, 388, 389, 391 and 787: "Figure 11" revised to "Figure 12"

Line 405, 410, 411 and 793: "Figure 12" revised to "Figure 13"

Line 407, 408, 409 and 799: "Figure 13" revised to "Figure 14"

Line 407, 412, 409 and 805: "Figure 14" revised to "Figure 15"

Line 421 and 832: "Table 2" revised to "Table 4"

Line 428, 430 and 811: "Figure 15" revised to "Figure 16"

Line 433-434,: "to evaluate the model capability of seasonal prediction of sea ice" added

Line 482,: "sophisticated sea ice albedo formulation" added

Line 518-520: "Bailey, D. A., Holland, M. M., DuVivier, A. K., Hunke, E. C., and Turner, A. K.: Impact of a New Sea Ice Thermodynamic Formulation in the CESM2 Sea Ice Component, Journal of Advances in Modeling Earth Systems, 12, e2020MS002154, https://doi.org/10.1029/2020MS002154, 2020." added

Line 547-548: "Daru, V., and Tenaud, C.: High order one-step monotonicity-preserving schemes for unsteady compressible flow calculations, Journal of Computational Physics, 193, 563-594, 2004." added

Line 578-579: "Jackett, D. R., and McDougall, T. J.: Minimal Adjustment of Hydrographic Profiles to Achieve Static Stability, Journal of Atmospheric and Oceanic Technology, 12, 381-389, 10.1175/1520-0426(1995)012<0381:MAOHPT>2.0.CO;2, 1995." added

Line 695-697: "Yang, C.-Y., Liu, J., and Xu, S.: Seasonal Arctic Sea Ice Prediction Using a Newly Developed Fully Coupled Regional Model With the Assimilation of Satellite Sea Ice Observations, Journal of Advances in Modeling Earth Systems, 12, e2019MS001938, https://doi.org/10.1029/2019MS001938, 2020" added

Line 700-701: "Zhang, J., and Hibler III, W. D.: On an efficient numerical method for modeling sea ice dynamics, Journal of Geophysical Research: Oceans, 102, 8691-8702, https://doi.org/10.1029/96JC03744, 1997." added

Line 713: Figure2 added;

Line 714-717: "Figure 2:   Concurrent mode of the coupled model. The small blocks under OCN or above ATM are the small subdomains in each node; the block under CPL

is the coupler. The red curve arrows indicate that the component models are sending data to the coupler and the red straight arrows indicate that the component models are reading data from the coupler. The horizontal arrows in the wall time indicate the time axis of each component model and the ticks on the time axis indicate the coupling time steps." added

Line 815-819: Table1 added;

Line 828-829: Table3 added;

[revised manuscript text omitted]

---

## Author Response (AR3)

**Reply to the Comments from Topical Editor**

*Comments to the Author:*

*Dear Dr. Ren and colleagues,*

*In concordance with the reviewers, I am excited to accept your paper for publication in GMD subject to minor grammatical and textual edits. I have attached a revision that I have done (apologies for the writing by hand, there were too many small changes to list them) which should address the reviewer's main concern about clarity in the writing. Please address these (please let me know if you can't read anything that I have added) and I'll be happy to send your paper off for type-setting.*

Reply:

Dear Alex Robel,

Thanks for your efforts on grammatical and text modification which make our paper more readable. We have revised the manuscript thoroughly according to your revision point by point.

Yours sincerely,

Shihe Ren

[revised manuscript text omitted]